# Almost Horizon-Free Structure-Aware Best Policy Identification with a Generative Model

**Andrea Zanette**
Institute for Computational and Mathematical Engineering,
Stanford University, CA
zanette@stanford.edu

**Mykel J. Kochenderfer**
Department of Aeronautics and Astronautics,
Stanford University, CA
mykel@stanford.edu

**Emma Brunskill**
Department of Computer Science,
Stanford University, CA
ebrun@cs.stanford.edu

## Abstract

This paper focuses on the problem of computing an $\epsilon$-optimal policy in a discounted Markov Decision Process (MDP) provided that we can access the reward and transition function through a generative model. We propose an algorithm that is initially agnostic to the MDP but that can leverage the specific MDP structure, expressed in terms of variances of the rewards and next-state value function, and gaps in the optimal action-value function to reduce the sample complexity needed to find a good policy, precisely highlighting the contribution of each state-action pair to the final sample complexity. A key feature of our analysis is that it removes all horizon dependencies in the sample complexity of suboptimal actions except for the intrinsic scaling of the value function and a constant additive term.

## 1 Introduction

A key goal is to design reinforcement learning (RL) agents that can leverage problem structure to efficiently learn a good policy, especially in problems with very long time horizons. Ideally the RL algorithm should be able to adjust without apriori information about the problem structure. Formal analyses that characterize the performance of such algorithms yielding instance-dependent bound help to advance our core understanding of the characteristics that govern the hardness of learning to make good decisions under uncertainty.

Though there is relatively limited work in reinforcement learning, strong problem-dependent guarantees are available for multi-armed bandits. In particular, well known bounds for online learning scale as a function of the gap between the expected reward of a particular action and the optimal action [ABF02] and also on the variance of the rewards [AMS09]. In the pure exploration setting in bandits, which is related to the setting we consider in this paper, there exist multiple algorithms with problem-dependent bounds [EMM06; MM94; MSA08; Jam+14; BMS09; ABM10; GGL12; KKS13] of this form. Ideally the complexity of learning to make good decisions in reinforcement learning tasks would scale with previously identified quantities of gap and variance over the next value function. As a step towards this, in this paper we introduce an algorithm for an RL agent operating in a discrete state and action space that has access to a generative model and can leverage problem-dependent structure to have strong instance-dependent PAC sample complexity bounds that are a function of the variance of the rewards and next state value functions, as well as the gaps between the optimal and suboptimal state-action values. While the sequential setting brings additional difficulties due to possibly long horizon, our bounds also show that in the dominant terms, our

approach avoids suffering any horizon dependence for suboptimal actions beyond the scaling of the value function. This significantly improves in statistical efficiency over prior worst-case bounds for the generative model case [GAMK13; Sid+18] and matches existing worst-case bounds in worst-case settings.

To do so we introduce a novel algorithm structure that acquires samples of state-action pairs in iterative rounds. A slight variant of the well known simulation lemma (see e.g. [KMN02]) suggests that in order to improve our estimate of the optimal value function and policy, it is sufficient to ensure that after each round of sampling, the confidence intervals shrink over the MDP parameter estimates of both the state–action pairs *visited by the optimal policy* and the state–action pairs *visited by the empirically-optimal policy*. While of course both are unknown, we show that we can implicitly maintain a set of candidate policies that are $\epsilon$-accurate, and by ensuring that we shrink the confidence sets of all state–action pairs likely to be visited by any such policy, we are also guaranteed (with high probability) to shrink the confidence intervals of the optimal policy. Interestingly we can show that by focusing on such state–action pairs, we can avoid the horizon dependence on suboptimal actions. The key idea is to take into account the MDP learned dynamics to enforce a constraint on the suboptimality of the candidate policies. The sampling strategy is derived by solving a minimax problem that minimizes the number of samples to guarantee that every policy in the set of candidate policies is accurately estimated. Importantly, this minimax problem can be reformulated as a convex minimization problem that can be solved with any standard solver for convex optimization.

Our algorithmic approach is quite different from many prior approaches, both in the generative model setting and the online setting. When a generative model is available, the available worst-case optimal algorithms [AMK12; Sid+18] allocate samples uniformly to all state and action pairs. We show our approach can be substantialy more effective for general case of MDPs with heterogeneous structure, and even for the pathologically hard instances because of the reduced horizon dependence on suboptimal actions. Note too that our approach is quite different from online RL algorithms that often (implicitly) allocate exploration budget to state-action pairs encountered by the policy with the most optimistic upper bound [JOA10; AOM17; OVR13; DB15; DLB17; SLL09; LH14], since here we explicitly reason about the reduction in the confidence intervals across a large set of policies whose value is near the *empirical* optimal value at this round.

## 2 Notation and Preliminaries

We consider discounted infinite horizon MDPs [SB18], which are defined by a tuple $\mathcal{M} = \langle \mathcal{S}, \mathcal{A}, p, r, \gamma \rangle$, where $\mathcal{S}$ and $\mathcal{A}$ are the state and action spaces with cardinality $S$ and $A$, respectively. We denote by $p(s' \mid s, a)$ the probability of transitioning to state $s'$ after taking action $a$ in state $s$ while $r(s, a) \in [0, 1]$ is the average instantaneous reward collected and $R(s, a) \in [0, 1]$ the corresponding random variable. The vector value function of policy $\pi$ is denoted with $V^\pi$. If $\rho$ is the initial starting distribution then $V(\rho) = \sum_s \rho_s V(s)$. The value function of the optimal policy $\pi^\star$ is denoted with $V^\star = V^{\pi^\star}$. We call $\mathrm{Var}\, R(s, a)$ and $\mathrm{Var}_{p(s,a)} V^\star$ the variance of $R(s, a)$ and of $V^\star(s')$ where $s' \sim p(s, a)$. The agent interacts with the MDP via a generative model that takes as input a $(s, a)$ pair and returns a random sample of the reward $R(s, a)$ and a random next state $s^+$ according to the transition model $s^+ \sim p(s, a)$. The reinforcement learning agent maintains an empirical MDP $\widehat{\mathcal{M}}_k = \langle \mathcal{S}, \mathcal{A}, \widehat{p}_k, \widehat{r}_k, \gamma \rangle$ for every iteration/episode $k$, and the maximum likelihood transition $\widehat{p}_k(s, a)$ and rewards $\widehat{r}_k(s, a)$ have received $n_{sa}^k$ samples. The $\widehat{V}_k^\star$ is the empirical estimate using MDP $\widehat{\mathcal{M}}_k$ of the empirical optimal policy $\widehat{\pi}_k^\star$. Variables with a hat refer to the empirical MDP $\widehat{\mathcal{M}}_k$ and the subscript indicates what iteration/episode they refer to. We denote with $\overline{w}_{sa}^{\pi,\rho} = \sum_{t=0}^\infty \gamma^t \Pr(s, a, t, \rho)$ the discounted sum of visit probabilities $\Pr(s, a, t, \rho)$ to the $(s, a)$ pair in timestep $t$ if the starting state is drawn uniformly from $\rho$ and $\widehat{\overline{w}}_{sa}^{\pi,k,\rho}$ is its analogous on $\widehat{\mathcal{M}}_k$. We use the $\tilde{O}(\cdot)$ notation to indicate a quantity that depends on $(\cdot)$ up to a $\mathrm{polylog}$ expression of a quantity at most polynomial in $S, A, \frac{1}{1-\gamma} \frac{1}{\delta}$, where $\delta$ is the "failure probability". Before proceeding, we first recall the following lemma from [GAMK13]:

**Lemma 2** (Simulation Lemma for Optimal Value Function Estimate [GAMK13])**.** *With probability at least* $1 - \delta$, *outside the failure event for any starting distribution* $\rho$ *it holds that:*

$$(V^\star - \widehat{V}_k^\star)(\rho) \leq \sum_{(s,a)} \widehat{\overline{w}}_{sa}^{\pi^\star,k,\rho} \left( (r - \widehat{r}_k)(s,a) + \gamma(p - \widehat{p}_k)(s,a)^\top V^\star \right) \leq \sum_{(s,a)} \widehat{\overline{w}}_{sa}^{\pi^\star,k,\rho} CI_{sa}(n_{sa}^k)$$

$$(V^\star - \widehat{V}_k^\star)(\rho) \geq \sum_{(s,a)} \widehat{\overline{w}}_{sa}^{\widehat{\pi}_k^\star,k,\rho} \left( (r - \widehat{r}_k)(s,a) + \gamma(p - \widehat{p}_k)(s,a)^\top V^\star \right) \geq - \sum_{(s,a)} \widehat{\overline{w}}_{sa}^{\widehat{\pi}_k^\star,k,\rho} CI_{sa}(n_{sa}^k)$$

The $CI_{sa}(n_{sa}^k)$ are Bernstein's confidence intervals (defined in more details in appendix A) after $n_{sa}^k$ samples over the rewards and transitions and are function of the unknown rewards and transition variances. The proof (see appendix) is a slight variation of lemma 3 in [GAMK13].

## 3 Sampling Strategy Given an Empirical MDP

We first describe how our approach will allocate samples to state–action pairs given a current empirical MDP, before presenting in the next section our full algorithm.

Lemma 1 suggests that to estimate the optimal value function it suffices to accurately estimate the $(s,a)$ pairs in the trajectories identified by two policies, namely the optimal policy $\pi^\star$ (optimal on $\mathcal{M}$) and the empirical optimal policy $\widehat{\pi}_k^\star$ (optimal on $\widehat{\mathcal{M}}_k$). Since $\pi^\star$ and $\widehat{\pi}_k^\star$ are unknown (in particular, $\widehat{\pi}_k^\star$ is a random variable prior to sampling), a common strategy is to allocate an identical number of samples uniformly [GAMK13; Sid+18] so that the confidence intervals $CI_{sa}(n_{sa}^k)$ are sufficiently small for all state–action pairs leading to a small $|(V^\star - \widehat{V}_k^\star)(\rho)|$; from here it is possible to show that the empirical optimal policy $\widehat{\pi}_k^\star$ can be extracted and that $|(V^\star - V^{\widehat{\pi}_k^\star})(\rho)|$ is also small (so $\widehat{\pi}_k^\star$ is near-optimal). Therefore, in the main text we mostly focus on showing that $|(V^\star - \widehat{V}_k^\star)(\rho)|$ is small, and defer additional details to the appendix. The proposed approach is to proceed in iterations or episodes. In each episode our algorithm implicitly maintains a set of candidate policies, which are near-optimal, and allocates more samples to the $(s,a)$ pairs visited by these policies to refine their estimated value. On the next episode those policies that are too suboptimal relative to their estimation accuracy are implicitly discarded. In particular, the samples are placed in a way that is related to the visit probabilities of the near-optimal empirical policies in addition to the variances of the reward and transitions of state–action pairs encountered in potentially good policies.

### 3.1 Oracle Minimax Program

Suppose we have already allocated some samples and have computed the maximum likelihood MDP $\widehat{\mathcal{M}}_k$ with empirical optimal policy $\widehat{\pi}_k^\star$ and know that the optimal value function estimate is at least $\epsilon_k$-accurate, i.e., $\|V^\star - \widehat{V}_k^\star\|_\infty \leq \epsilon_k$. How should we allocate further sampling resources to improve the accuracy in the optimal value function estimate? The idea is given by the simulation lemma (lemma 2): in order to see an improvement after sampling (i.e., in the next episode $k+1$) the maximum likelihood MDP $\widehat{\mathcal{M}}_{k+1}$ must have smaller confidence intervals $CI_{sa}(n_{sa}^{k+1})$ in the $(s,a)$ pairs visited by $\pi^\star$ and the empirical optimal policy $\widehat{\pi}_{k+1}^\star$ on $\widehat{\mathcal{M}}_{k+1}$. Both are of course unknown. However, we introduce the constraint $(\widehat{V}_k^\star - \widehat{V}_k^\pi)(\rho) \leq C\epsilon_k$ that restricts sampling to $C\epsilon_k$-optimal policies (and starting distributions) on $\widehat{\mathcal{M}}_k$. Here, $C$ is a numerical constant that *will ensure that $\pi^\star$ and $\widehat{\pi}_{k+1}^\star$ satisfy this condition and are therefore allocated enough samples.* Given $C$ and $\epsilon_k$, the idea is that we should choose a sampling strategy $\{n_{sa}\}_{sa}$ with high enough samples to ensure $\sum_{(s,a)} \widehat{\overline{w}}^{\pi,k+1,\rho} CI_{sa}(n_{sa}^{k+1}) \leq \epsilon_{k+1}$ for all policies that satisfy $(\widehat{V}_k^\star - \widehat{V}_k^\pi)(\rho) \leq C\epsilon_k$ so that Lemma 2 ensures $|(V^\star - \widehat{V}_{k+1}^\star)(\rho)| \leq \epsilon_{k+1} = \epsilon_k/2$. This is equivalent to solving the following[1]:

**Definition 1** (Oracle Minimax Problem)**.**

$$\min_n \max_{\pi,\rho} \quad \sum_{(s,a)} \widehat{\overline{w}}_{sa}^{\pi,k+1,\rho} CI_{sa}(n_{sa}^{k+1}), \quad s.t. \quad (\widehat{V}_k^\star - \widehat{V}_k^\pi)(\rho) \leq C\epsilon_k, \sum_{(s,a)} n_{sa} \leq n_{max}. \quad (1)$$

[1]For space, we omit the constraint $\rho_s \geq 0$ and $\|\rho\|_1 = 1$ on the starting distribution.

Here the vector of the discounted sum of visit probabilities $\widehat{\overline{w}}^{\pi,k+1,\rho}$ is computable from the linear system $(I - \gamma(\widehat{P}_{k+1}^\pi)^\top)\widehat{\overline{w}}^{\pi,k+1,\rho} = \rho$ and $n_{max}$ is a guess on the number of samples needed to ensure that the objective function is $\leq \epsilon_k/2$. We call this problem the *oracle minimax problem* because it uses the next-episode empirical visit probabilities $\widehat{\overline{w}}^{\pi,k+1,\rho}$ which are not known. In addition, it uses the true variance of the next state value function (embedded in the definition of confidence intervals $CI_{sa}(n_{sa}^k)$). As these quantities are unknown in episode $k$, the program cannot be solved.

### 3.2 Algorithm Minimax Program

This section shows how to construct a minimax program that is 'close' enough to the Oracle minimax problem (Equation 1) but is function of only empirical quantities computable from $\widehat{\mathcal{M}}_k$. The idea is 1) to avoid using the next-episode empirical distribution $\widehat{\overline{w}}^{\pi,k+1,\rho}$ and instead use the currently-computable $\widehat{\overline{w}}^{\pi,k,\rho}$ and 2) use the empirical variance of the next state value function $\mathrm{Var}_{\widehat{p}_k(s,a)}\widehat{V}_k^\star$ instead of the real, unknown variance $\mathrm{Var}_{p(s,a)}V^\star$. Estimating the visit distribution $\widehat{\overline{w}}^{\pi,k+1,\rho}$ accurately leads to a high sample complexity; fortunately we are able to claim that the product between *the visit distribution shift* $\widehat{\overline{w}}^{\pi,k+1,\rho} - \widehat{\overline{w}}^{\pi,k,\rho}$ *and the confidence interval vector* $CI^{k+1}$ *on the rewards and transitions* is already small if policy $\pi$ has received enough samples along its trajectories before the current episode. Let us rewrite the objective function of equation 1 as a function of the visit distribution on $\widehat{\mathcal{M}}_k$ plus a term that takes into account the shift in the distribution from $\widehat{\mathcal{M}}_k$ to $\widehat{\mathcal{M}}_{k+1}$:

$$\sum_{(s,a)} \widehat{\overline{w}}_{sa}^{\pi,k+1,\rho} CI_{sa}(n_{sa}^{k+1}) = \sum_{(s,a)} \underbrace{\widehat{\overline{w}}_{sa}^{\pi,k,\rho}}_{\text{Computable}} CI_{sa}(n_{sa}^{k+1}) + \sum_{(s,a)} \underbrace{(\widehat{\overline{w}}_{sa}^{\pi,k+1,\rho} - \widehat{\overline{w}}_{sa}^{\pi,k,\rho})}_{\text{Shift of Empirical Distributions}} CI_{sa}(n_{sa}^{k+1})$$

Lemma 9 in appendix allows us to claim that the rightmost summation above is less than $2Cp(n_{min})\epsilon_k$ for both[2] $\pi = \pi^\star$ and $\widehat{\pi}_{k+1}^\star$. Here $Cp(n_{min})$ is defined in appendix A and can be made (see lemma 16) for example $< 1/100$ by allocating a small constant number of samples $\tilde{O}(S/(1-\gamma)^2)$ to each $(s,a)$ pair[3], independent of the desired accuracy $\epsilon_{k+1}$. This way we can ensure that we can use $\widehat{\overline{w}}^{\pi,k,\rho}$ instead of $\widehat{\overline{w}}^{\pi,k+1,\rho}$ plus a small correction term $\ll \epsilon_k$.

Now the only quantities that are not known by the algorithm are the variance of the transitions and rewards that appear in the confidence intervals $CI_{sa}(n_{sa}^{k+1})$. Precisely, to estimate the variance of the transitions $\mathrm{Var}_{p(s,a)}V^\star$ in the $(s,a)$ pair, we need to known both the transition probability $p(s,a)$ and the true value function $V^\star$, both of which are unknown. Fortunately it is possible to use the empirical transitions $\widehat{p}_k(s,a)$ and the empirical value function $\widehat{V}_k^\star$ plus a fast-shrinking (as a function of the number of samples) correction term. Since this analysis was similarly performed in prior papers for this setting [GAMK13; Sid+18], we defer its discussion to Lemma 11 in the appendix. With these corrections ($B_{ksa}$, defined in appendix A, is the variance correction and $2\epsilon_k/625$ accounts for the distribution shift) we can write the following minimax problem:

**Definition 2** (Algorithm Minimax Problem).

$$\min_n \max_{\pi,\rho} \quad \sum_{(s,a)} \widehat{\overline{w}}_{sa}^{\pi,k,\rho} (\widehat{CI}_{sa}(n_{sa}^{k+1}) + B_{ksa}) + 2\epsilon_k/625$$

$$\text{s.t.:} \quad (\widehat{V}_k^\star - \widehat{V}_k^\pi)(\rho) \leq C\epsilon_k; \quad \sum_{(s,a)} n_{sa} \leq n_{max}; \quad (I - \gamma(\widehat{P}_k^\pi)^\top)\widehat{\overline{w}}^{\pi,k,\rho} = \rho. \qquad (2)$$

Here $\widehat{CI}_{sa}(n_{sa}^{k+1})$ are the confidence intervals evaluated with the empirical variances defined in Appendix A. This program is fully expressed in terms of empirical quantities that depends on $\widehat{\mathcal{M}}_k$. As long as a solution to the above minimax program is $\leq \epsilon_k/2$ the oracle objective function will also

be $\leq \epsilon_k/2$ at the solution of the program (for more details see Lemma 6 in the Appendix). In other words, by solving the minimax program (def 2) we put enough samples to satisfy the oracle program 1, which ensures accuracy in the value function estimate through Lemma 2.

## 4 Algorithm

We now take the sampling approach described in the previous section and use it to construct an iterative algorithm for quickly learning a near-optimal or optimal policy given access to a generative model. Specifically we present *BESt POlicy identification with no Knowledge of the Environment* (BESPOKE) in Algorithm 1. The algorithm proceeds in episodes. Each episode starts with an empirical MDP $\widehat{\mathcal{M}}_k$ whose optimal value function $\widehat{V}_k^\star$ is $\epsilon_k$ accurate $\|V^\star - \widehat{V}_k^\star\|_\infty \leq \epsilon_k$ under an inductive assumption. The sam-

---

**Algorithm 1** BESPOKE

**Input:** Failure probability $\delta > 0$, accuracy $\epsilon_{Input} > 0$
Set $\epsilon_1 = \frac{1}{1-\gamma}$ and allocate $n_{min}$ samples to each $(s,a)$ pair
**for** $k = 1, 2, \ldots$
    **for** $n_{max} = 2^0, 2^1, 2^2, \ldots$
        Solve the optimization program of definition 7 (appendix)
        **if** the optimal value of the program of definition 7 is $\leq \frac{\epsilon_k}{2}$
            Break and return sampling strategy $\{n_{sa}^{k+1}\}_{sa}$
    Query the generative model up to $n_{sa}^{k+1}, \forall(s,a)$
    Compute, $\widehat{\pi}_{k+1}^\star$ and $\widehat{V}_{k+1}^\star$
    Set $\epsilon_{k+1} = \frac{\epsilon_k}{2}$
    **if** $\epsilon_{k+1} \leq \epsilon_{Input}$
        Break and return the policy $\widehat{\pi}_{k+1}^\star$

---

ples are allocated at each episode $k$ by solving an optimization program equivalent to that in definition 2 to halve the accuracy in the value function estimate, i.e., $\|V^\star - \widehat{V}_{k+1}^\star\|_\infty \leq \epsilon_{k+1} = \epsilon_k/2$. In the innermost loop of the algorithm the required number of samples for the next episode is guessed $n_{max} = 1, 2, 4, 8, \ldots$, until $n_{max}$ is sufficient to ensure that the objective function of the minimax problem of definition 2 will be $\leq \epsilon_k/2$; the purpose of the inner loop is to avoid putting more samples than needed and allows us to obtain the sample complexity result of Theorem 2. In Appendix G we reformulate the optimization program 2 (described more precisely in Definition 5 in the appendix) obtaining a convex minimization program that avoids optimizing over the policy and instead works directly with the distribution $\widehat{\overline{w}}^{\pi,k,\rho}$; this can be efficiently solved with standard techniques from convex optimization [BV04].

**Theorem 1** (BESPOKE Works as Intended). *With probability at least $1 - \delta$, in every episode $k$ BESPOKE maintains an empirical MDP $\widehat{\mathcal{M}}_k$ such that its optimal value function $\widehat{V}_k^\star$ and its optimal policy $\widehat{\pi}_k^\star$ satisfy:*

$$\|V^\star - \widehat{V}_k^\star\|_\infty \leq \epsilon_k, \quad \|V^\star - V^{\widehat{\pi}_k^\star}\|_\infty \leq 2\epsilon_k$$

*where $\epsilon_{k+1} \stackrel{def}{=} \frac{\epsilon_k}{2}, \forall k$. In particular, when BESPOKE terminates in episode $k_{Final}$ it holds that $\frac{\epsilon_{Input}}{2} \leq \epsilon_{k_{Final}} \leq \epsilon_{Input}$.*

The proof is reported in the appendix, and shows by induction that for every episode $k$, $\pi^\star$ and $\widehat{\pi}_{k+1}^\star$ are in the set of 'candidate' policies because they are near-optimal on $\widehat{\mathcal{M}}_k$, satisfying $(\widehat{V}_k^\star - \widehat{V}_k^{\pi^\star})(\rho) \leq C\epsilon_k$ and $(\widehat{V}_k^\star - \widehat{V}_k^{\widehat{\pi}_{k+1}^\star})(\rho) \leq C\epsilon_k$ for all $\rho$ and are therefore allocated enough samples; this guarantees accuracy in $\widehat{V}_{k+1}^\star$ by Lemma 2.

## 5 Sample Complexity Analysis

To analyze the sample complexity of BESPOKE we derive another optimization program function of only problem dependent quantities. We 1) shift from the empirical visit distribution $\widehat{\overline{w}}^{\pi,k,\rho}$ on $\widehat{\mathcal{M}}_k$ to the "real" visit distribution $\overline{w}^{\pi,\rho}$ on $\mathcal{M}$; 2) move from empirical confidence intervals to those evaluated with the true variances; and finally 3) relax the near-optimality constraint on the policies by using Lemma 7 in the appendix (for an appropriate numerical constant $C^\star > C$) in order to be able to use the value functions on $\mathcal{M}$:

$$(\widehat{V}_k^\star - \widehat{V}_k^\pi)(\rho) \leq C\epsilon_k \rightarrow (V^\star - V^\pi)(\rho) \leq C^\star \epsilon_k, \quad \forall \rho. \tag{3}$$

In the end, we have enlarged the feasible set of the algorithm minimax problem while upper bounding its objective function obtaining:[4]

**Definition 3** ($\star$-Minimax Program).

$$\min_n \max_{\overline{w}^{\pi,\rho}} \sum_{(s,a)} \overline{w}_{sa}^{\pi,\rho}(CI_{sa}(n_{sa}^{k+1}) + 2B_{ksa}) + \epsilon_k/25 \qquad (4)$$

subject to the constraints ($r \in \mathbb{R}^{SA}$ is the reward vector):

$$\underbrace{(V^\star - V^\pi)(\rho)}_{V^\star(\rho) - (\overline{w}^{\pi,\rho})^\top r} \leq C^\star \epsilon_k; \quad \sum_{(s,a)} n_{sa} \leq n_{max}; \quad (I - \gamma(P^\pi)^\top)\overline{w}^{\pi,\rho} = \rho. \qquad (5)$$

This is made rigorous in Lemma 6, but essentially we have obtained a minimax program whose solution can be studied in terms of problem dependent quantities; in particular, its solution in terms of number of samples $n_{sa}$ upper bounds the sample complexity of the algorithm in every episode.

**Problem Dependent Analysis**  Due to space constraints, here we sketch the sample complexity analysis of suboptimal actions to make the gaps $\Delta_{sa} \overset{def}{=} V^\star(s) - Q^\star(s,a)$ appear while simultaneously eliminating the horizon dependence. We recall the following (e.g., Lemma 5.2.1 in [Kak+03]; see also our appendix):

**Lemma 1** (Sum of Losses). *It holds that:*

$$(V^\star - V^\pi)(\rho) = \sum_{(s,a)} \overline{w}_{sa}^{\pi,\rho}(\underbrace{Q^\star(s,\pi^\star(s)) - Q^\star(s,a)}_{\overset{def}{=} \Delta_{sa}}) = \sum_{(s,a)} \overline{w}_{sa}^{\pi,\rho}\Delta_{sa} \qquad (20)$$

Lemma 1 expresses the value of a suboptimal policy as a sum of per-step losses $\Delta_{sa}$ weighted by the discounted sum of probabilities of being in that $(s,a)$ pair. The key step that enables us to obtain strong problem dependent bounds and to remove the horizon dependence for suboptimal actions is synthesized in the following short lemma, where we ignore the term $(\sum_{(s,a)} 2\overline{w}_{sa}^{\pi,\rho}B_{ksa} + 3\epsilon_k/625)$.

**Lemma 1** (Gap-Confidence Interval Lemma). *If $(\pi,\rho)$ satisfies $(V^\star - V^\pi)(\rho) \leq C^\star\epsilon_k$ then a sample complexity:*

$$n_{sa} = \tilde{O}\left(\underbrace{\frac{\mathrm{Var}\,R(s,a)}{\Delta_{sa}^2} + \frac{1}{\Delta_{sa}}}_{\text{Reward Estimation}} + \underbrace{\frac{\gamma^2\,\mathrm{Var}_{p(s,a)}\,V^\star}{\Delta_{sa}^2} + \frac{\gamma}{(1-\gamma)\Delta_{sa}}}_{\text{Transition Estimation}}\right), \quad \forall(s,a) \qquad (6)$$

*suffices to ensure*

$$\max_{\overline{w}^{\pi,\rho}} \sum_{(s,a)} \overline{w}_{sa}^{\pi,\rho}CI_{sa}(n_{sa}^{k+1}) \leq \frac{\epsilon_k}{2}. \qquad (7)$$

*Proof.* A direct computation shows that if $n_{sa}^{k+1}$ satisfies equation 6 with appropriate constants[5] then:

$$CI_{sa}(n_{sa}^{k+1}) \leq \frac{\Delta_{sa}}{2C^\star}. \qquad (8)$$

This justifies the first inequality below:

$$\sum_{(s,a)} \overline{w}_{sa}^{\pi,\rho}CI_{sa}(n_{sa}^{k+1}) \leq \frac{1}{2C^\star}\sum_{(s,a)} \overline{w}_{sa}^{\pi,\rho}\Delta_{sa} = \frac{1}{2C^\star}(V^\star - V^\pi)(\rho) \leq \frac{1}{2}\epsilon_k. \qquad (9)$$

The equality follows from lemma 1 and the last inequality from the constraint on the optimality of $\pi$. $\qquad \square$

They key idea is that by *having confidence intervals of the same size as the gaps is sufficient to estimate the policy as accurately as its suboptimality gap* $(V^\star - V^\pi)(\rho)$, regardless of the horizon. By augmenting this argument with the law of total variance [GAMK13], splitting into further subcases, and by taking into account the correction terms we obtain:

**Theorem 2** (Sample Complexity of the Algorithm BESPOKE). *With probability at least $1 - \delta$, the total sample complexity of* BESPOKE *up to episode $k$ is upper bounded by $\sum_{(s,a)} n_{sa}$ where $n_{sa}$ is the total number of samples allocated to the $(s,a)$ pair:*

$$n_{sa} = \tilde{O} \left( \min \left\{ \frac{1}{(1-\gamma)^3(\epsilon_k)^2}, \frac{\operatorname{Var} R(s,a) + \gamma^2 \operatorname{Var}_{p(s,a)} V^\star}{(1-\gamma)^2(\epsilon_k)^2} + \frac{1}{(1-\gamma)^2(\epsilon_k)}, \right. \right. \tag{166}$$

$$\left. \left. \frac{\operatorname{Var} R(s,a) + \gamma^2 \operatorname{Var}_{p(s,a)} V^\star}{\Delta_{s,a}^2} + \frac{1}{(1-\gamma)\Delta_{s,a}} \right\} + \frac{\gamma S}{(1-\gamma)^2} \right). \tag{167}$$

Notice that the BESPOKE would suffer a worst-case sample complexity similar to [GAMK13; Sid+18] only in the initial phases of learning, i.e., whenever $\epsilon_k$ is much larger than the gaps.

# 6  Significance of the Bound

We motivate the importance of theorem 2 by specializing the result in two noteworthy cases.

**Sample Complexity to Identify the Best Policy and the Worst-Case Lower Bound**   If the optimal policy is unique, define the minimum gap $\Delta_{min} = \min_{s,a,a\neq\pi^\star(a)} \Delta_{sa}$. To identify the optimal policy we must set $\epsilon_{Input} \leq \Delta_{min}$ and the sample complexity of BESPOKE at termination becomes:

$$\tilde{O} \left( \underbrace{\sum_s \min \left\{ \frac{1}{(1-\gamma)^3 \Delta_{min}^2}, \frac{\operatorname{Var} R(s,\pi^\star(s)) + \gamma^2 \operatorname{Var}_{p(s,\pi^\star(s))} V^\star}{(1-\gamma)^2 \Delta_{min}^2} + \frac{1}{(1-\gamma)^2 \Delta_{min}} \right\}}_{\text{ESTIMATING } \pi^\star} \right.$$

$$\left. + \underbrace{\sum_{(s,a)|a\neq\pi^\star(s)} \left( \frac{\operatorname{Var} R(s,a) + \gamma^2 \operatorname{Var}_{p(s,a)} V^\star}{\Delta_{sa}^2} + \frac{1}{(1-\gamma)\Delta_{sa}} \right)}_{\text{RULING-OUT SUBOPTIMAL ACTIONS}} + \underbrace{\frac{\gamma S^2 A}{(1-\gamma)^2}}_{\text{CONSTANT}} \right) \tag{10}$$

One of our core contributions is that we suffer a dependence on the horizon $1/(1 - \gamma)$ only in estimating the optimal $(s,a)$ pairs (first summation over the state space). *The summation over suboptimal $(s,a)$ is independent of the horizon*, although of the horizon implicitly affects the scaling of the variance $\operatorname{Var}_{p(s,a)} V^\star$ and explicitly the maximum value function range (term $1/(1-\gamma)\Delta_{sa}$).

It is important to compare the above result with the established lower bound [GAMK13] which is $\Omega(\frac{N}{(1-\gamma)^3 \epsilon^2})$ to obtain an $\epsilon$-accurate policy, where $N$ is the number of state-action pairs. Since $\Delta_{sa} = \Delta_{min}$, $\forall (s,a), a \neq \pi^\star(s)$ in the lower bound construction and the variance is maximum $\operatorname{Var}_{p(s,a)} V^\star \leq 1/(1-\gamma)^2$, we are able to identify the optimal policy in $\tilde{O} \left( \frac{S}{(1-\gamma)^3 \Delta_{min}^2} + \frac{S(A-1)}{(1-\gamma)^2 \Delta_{min}^2} + \frac{S^2 A}{(1-\gamma)^2} \right)$ samples which improves[6] on the worst case bound $\tilde{O} \left( \frac{SA}{(1-\gamma)^3 \Delta_{min}^2} + \frac{S^2 A}{(1-\gamma)^2} \right)$ of [GAMK13; Sid+18] by a full horizon factor for suboptimal actions. While our result can be surprising at first, it does not contradict the lower bound: the lower bound makes no attempt to distinguish between optimal and suboptimal actions as it is only expressed in terms of *total $(s,a)$ pairs $N$*, and the construction uses a number of $(s,a)$ pairs that is a *constant* multiple of the state space cardinality, i.e., one could only deduce $\Omega(\frac{S}{(1-\gamma)^3 \Delta_{min}^2})$ as a lower bound. Our result, therefore, *does not violate the lower bound*, but rather it shows that while we must suffer an unavoidable worst-case $1/(1-\gamma)^3$ factor on the state space corresponding to the optimal $(s,a)$ pairs, the dependence on the planning horizon is absent for suboptimal $(s,a)$ except for the scaling of the value function implicit in the variance. Surprisingly, excluding the constant term $\frac{S^2 A}{(1-\gamma)^2}$, suboptimal $(s,a)$ pairs get a combined number of samples

$\tilde{O}\left(\sum_{(s,a)|a\neq\pi^\star(s)}\left(\frac{\operatorname{Var}R(s,a)+\gamma^2\operatorname{Var}_{p(s,a)}V^\star}{\Delta_{sa}^2}+\frac{1}{(1-\gamma)\Delta_{sa}}\right)\right)$ which is the sample complexity (ignoring log and constant factors) that a variance-aware bandit algorithm for best arm identification would need (see e.g., [GGL12], appendix B) to 'reject' these suboptimal arms provided that it can obtain samples[7] of the random variable $R(s,a)+\gamma V^\star(s')$, $s'\sim p(s,a)$. In this case, however, the $V^\star$ vector would need to be known to the bandit algorithm. In other words, the sample complexity of BESPOKE at termination consists of two main terms: a leading order term with a dependence on the state space with an unavoidable (due to the lower bound) dependence on the horizon $\frac{1}{1-\gamma}$, and an horizon-free bandit-like sample complexity to rule out suboptimal actions as if the optimal value function $V^\star$ was known.

**BESPOKE applied to Bandits**   Finally, if $\gamma=0$ we are in the bandit setting, and the sample complexity of BESPOKE at step $k$ becomes exactly (since $\operatorname{Var}R(s,a)\le 1$):

$$\tilde{O}\left(\sum_{(s,a)}\left(\frac{\operatorname{Var}R(s,a)}{\max\{\epsilon_k^2,\Delta_{sa}^2\}}+\frac{1}{\max\{\epsilon_k,\Delta_{sa}\}}\right)\right)\le\tilde{O}\left(\sum_{(s,a)}\frac{1}{\max\{\epsilon_k^2,\Delta_{sa}^2\}}\right) \qquad (11)$$

This matches the best-known sample complexity bound for best arm identification for tabular bandit with gaps and variances [ABM10; GGL12] except for constants and log terms. This is encouraging as it suggests it may be possible to have algorithms with a smooth transition in sample complexity as a function of the discount factor when moving from a bandit to an RL setting.

## 7   Related Literature and Conclusion

**Related Literature**   In the more challenging setting of online exploration (i.e., without a generative model) the PAC literature [DB15; DLB17; LH14; SLL09] directly provides algorithms to identify an $\epsilon$-optimal policy with high probability in the worst-case. Gap-aware analyses exists, see for example [BK97; TB08; OPT18] for asymptotic regret bounds on ergodic MDPs with matching upper and lower bounds and with an emphasis on the minimum gap; since these analyses look at the asymptotic regret they are not comparable to the proposal here. Very recently [SJ19] presents a gap-based non-asymptotic regret bound for episodic MDPs but not yet free of the horizon and dependencies on $\Delta_{min}$. Gaps in MDPS have also been used to justify the observed relation between the value function accuracy and the resulting policy performance [FSM10]. In addition, [EMM06; Bru10] also propose an algorithm and PAC bounds that depend the minimum gap, but the results do not leverage recent advances in tighter sample complexity analysis. [JOA10] presents a regret bound based on the same quantity. The maximum variance of the next-state optimal value function is discussed in [MMM14; ZB19].

The closest related work in the PAC setting similarly assumes access to a generative model, and provides near-matching worst-case sample complexity upper and lower bounds [AMK12] for tabular MDPs even in terms of computational complexity [Sid+18]. However, this work focuses on near-optimal worst-case performance: as these algorithms allocate samples uniformly they do not adapt to the problem structure. Finally, [Aga+19] show how to improve on the constant sample complexity term for model based approaches like the one we use here; it is possible that their techniques can be applied to our setting.

**Conclusion**   This work leverages domain structure, notably the action-value function gaps, to eliminate the impact of the horizon when ruling out suboptimal actions to identify a near-optimal policy for discounted-reward Markov decision processes using a generative model, except for a constant term and the inherent value function scaling. This is achieved through a tractable algorithm. In doing so, our finite time sample complexity analysis quantifies the sample complexity contribution of each state-action pair as a function of the action-value function gaps and variances of the rewards and next-state value function, and recovers the best-known bounds (excepts for logs and constants) when deployed to bandit instances using these quantities.

Our work provides at least two important analytical tools: 1) the way we relate the suboptimality of the policies with the gaps to reduce the dependence on the horizon is new, and could be used in

other settings to make the gap appear while simultaneously reducing the horizon dependence 2) the way we analyze the visit distribution shift induced by the policies, weighted by the local reward and transition confidence intervals, and show it is small, is another analytical contribution of our work which can be extended to the settings where one is interested in obtaining a good policy from a given starting distribution $\rho$ as opposed to all starting states.

## Acknowledgment

This work is partially supported by a Total Innovation Fellowship program, an NSF CAREER award and an Office of Naval Research Young Investigator Award. The authors are grateful to the reviewers for the high-quality reviews and suggestions.

## Footnotes

[2]Lemma 9 bounds this term as $2Cp(n_{min})\epsilon_k^\pi$ for $\pi = \pi^\star, \pi = \widehat{\pi}_{k+1}^\star$, respectively; $\epsilon_k^\pi$ is defined in appendix A and represents the "accuracy" of policy $\pi$ in episode $k$. To ensure $\epsilon_k^\pi \leq \epsilon_k$ we need an inductive argument which is sketched out in the main theorem (Theorem 1).

[3]As we will shortly see, this will contribute only a constant term to the final sample complexity.

[4]The relaxed optimization program is over the distribution induced by the policy. Here, $P^\pi$ is the transition matrix identified by the policy $\pi$ on $\mathcal{M}$.

[5]Note that, in particular, $C^\star$ is a constant.

[6]The paper [Sid+18] has the same bound as [GAMK13] but avoids the constant term $\frac{S^2 A}{(1-\gamma)^2}$.

[7]Here, $\operatorname{Var}R(s,a)+\gamma^2\operatorname{Var}_{p(s,a)}V^\star$ is the variance of the random variable $R(s,a)+\gamma V^\star(s')$ with $s'\sim p(s,a)$. Note the scaling of this random variable, which has range $\frac{1}{1-\gamma}$.

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
