[Supplementary Material]

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

[8]For this passage we need $Cp(n_{min}) < 1$, but our choice of $n_{min}$ ensures that.

[9]Note that $j$ and $k$ are flipped in the two lemmas.

[10]the lemma ensures $|(V^\star - \widehat{V}_p^\star)(\rho)| \le \epsilon_p$ for all $\rho$, so choose $\rho$ to be the point mass on a starting state

[11]Notice that this part refers to policy $\widehat{\pi}_j^\star$ and not $\widehat{\pi}_{k+1}^\star$. This condition is ensured by the inductive hypothesis which holds in all episodes up to $k$.

[12]Note that $C^\star$ is just a constant.

[13]We drop all subscripts on $w$ in this section for simplicity.

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

# A  Symbols

Table 1: Notation

| | | |
|---:|:---:|:---|
| $\delta'_n$ | $\overset{def}{=}$ | $\frac{\delta}{4SAn^2}$ |
| $c_n$ | $\overset{def}{=}$ | $\ln(4/\delta'_n)$ |
| $\rho$ | $\overset{def}{=}$ | starting distribution |
| $\mathcal{M}$ | $\overset{def}{=}$ | true MDP |
| $\widehat{\mathcal{M}}_k$ | $\overset{def}{=}$ | empirical MDP at step $k$ |
| $V^\star$ | $\overset{def}{=}$ | optimal value function on $\mathcal{M}$ |
| $\widehat{V}_k^\star$ | $\overset{def}{=}$ | optimal value function on $\widehat{\mathcal{M}}_k$ |
| $\pi^\star$ | $\overset{def}{=}$ | optimal policy on $\mathcal{M}$ |
| $\widehat{\pi}_k^\star$ | $\overset{def}{=}$ | empirical optimal policy on $\widehat{\mathcal{M}}_k$ |
| $\overline{w}_{sa}^{\pi,\rho}$ | $\overset{def}{=}$ | visit probability to $(s,a)$ on $\mathcal{M}$ upon following $\pi$ with starting distribution $\rho$ |
| $\widehat{\overline{w}}_{sa}^{\pi,k,\rho}$ | $\overset{def}{=}$ | visit probability to $(s,a)$ on $\widehat{\mathcal{M}}_k$ upon following $\pi$ with starting distribution $\rho$ |
| $V(\rho)$ | $\overset{def}{=}$ | $\sum_i \rho_i V(s_i)$ |
| $\Delta_{sa}$ | $\overset{def}{=}$ | $Q^\star(s,\pi^\star(s)) - Q^\star(s,a)$ |
| $BI(\sigma, b, n)$ | $\overset{def}{=}$ | $\sqrt{\frac{2c_n\sigma^2}{n}} + \frac{bc_n}{3(n-1)}$ (Bernstein Inequality) |
| $CI_{sa}(n_{sa}^k)$ | $\overset{def}{=}$ | $BI(\mathrm{Var}\, R(s,a), 1, n_{sa}^k) + \gamma BI(\mathrm{Var}_{p(s,a)}\, V^\star, \frac{1}{(1-\gamma)}, n_{sa}^k)$ |
| $\widehat{CI}_{sa}(n_{sa}^k)$ | $\overset{def}{=}$ | $BI(\mathrm{Var}\, \widehat{R}(s,a), 1, n_{sa}^k) + \gamma BI(\mathrm{Var}_{\widehat{p}_k(s,a)}\, \widehat{V}_k^\star, \frac{1}{(1-\gamma)}, n_{sa}^k)$ |
| $CI^k$ | $\overset{def}{=}$ | Vector containing the $CI_{sa}(n_{sa}^k)$ values |
| $B_{ksa}$ | $\overset{def}{=}$ | $\frac{2c_n}{(1-\gamma)(n_{sa}-1)} + \gamma\sqrt{\frac{2c_n}{n_{sa}}}\epsilon_k$ |
| $B_k$ | $\overset{def}{=}$ | Vector containing the $B_{ksa}$ values |
| $\epsilon_k^\pi$ | $\overset{def}{=}$ | $\max_{\rho, \rho \geq 0, \|\rho\|_1 = 1} \sum_{(s,a)} \widehat{\overline{w}}_{sa}^{\pi,k,\rho} CI_{sa}(n_{sa}^k)$ |
| $C$ | $\overset{def}{=}$ | 20 |
| $C^\star$ | $\overset{def}{=}$ | $\frac{2+C}{1-Cp(n_{min})}$ |
| $Cp(n_{min})$ | $\overset{def}{=}$ | $\frac{\gamma}{(1-\gamma)}\sqrt{\frac{2Sc_n}{n_{min}}}$ |
| $n$ | $\overset{def}{=}$ | vector that contains the number of samples to each $(s,a)$, unless it's a generic scalar |
| $\widehat{S}_k$ | $\overset{def}{=}$ | $\{\pi \mid (\widehat{V}_k^\star - \widehat{V}_k^\pi)(\rho) \leq C\epsilon_k, \quad \forall \rho \geq 0, \|\rho\|_1 = 1\}$ |
| $S_k$ | $\overset{def}{=}$ | $\{\pi \mid (V^\star - V^\pi)(\rho) \leq C^\star\epsilon_k, \quad \forall \rho \geq 0, \|\rho\|_1 = 1\}$ |
| $n_{min}$ | $\overset{def}{=}$ | $\frac{2 \times 625^2 \gamma^2 Sc_n}{(1-\gamma)^2}$ |

# B   Standard Concentration Inequalities

**Proposition 1** (Bernstein's Inequality). *Let $X_1, \cdots, X_n$ be i.i.d. random variables with values in $[0, C]$ and let $\delta' > 0$. Then with probability at least $1 - \delta'$ in $(X_1, \ldots, X_n)$ we have:*

$$\left| \mathbb{E}\, X - \frac{1}{n} \sum_{i=1}^{n} X_i \right| \leq \sqrt{\frac{2 \operatorname{Var} X \ln (2/\delta')}{n}} + \frac{C \ln(2/\delta')}{3n}. \tag{12}$$

*Proof.* See [MP09], Theorem 3. □

**Corollary 1** (Bernstein's inequality applied to Transition Probabilities). *Let $p$ be a $k$-dimensional transition probability vector (such that $\|p\|_1 = 1$) and let $\widehat{p}$ be its maximum likelihood estimate. Let $\delta' > 0$. Then with probability at least $1 - \delta'$ we have:*

$$\left| \widehat{p}_i - p_i \right| \leq \sqrt{\frac{2p_i \ln (2k/\delta')}{n}} + \frac{2 \ln(2k/\delta')}{3n} \tag{13}$$

*where $p_i$ is the $i$-th component of $p$.*

*Proof.* Immediate from Bernstein's inequality with the variance $p_i(1 - p_i)$ of a Bernoulli random variable and a union bound over $k$. □

**Proposition 2** (Converge Rate of Empirical Variance). *Let $V^\star$ be a fixed vector with values in $[0, C]$ and let $\delta' > 0$:*

$$\left| \sqrt{\operatorname*{Var}_{\widehat{p}_k(s,a)} V^\star} - \sqrt{\operatorname*{Var}_{p(s,a)} V^\star} \right| \leq C \sqrt{\frac{2 \ln (2/\delta')}{n - 1}}. \tag{14}$$

*Proof.* See [MP09], Theorem 10. □

**Proposition 3** (Weissman et al. Inequality). *Let $\widehat{p}$ be the maximum likelihood probability vector of the distribution $p$ obtained by drawing i.i.d. samples from the discrete distribution $p$ with $k$ point masses, and let $\delta' > 0$. With probability at least $1 - \delta'$ it holds that:*

$$\|\widehat{p} - p\|_1 \leq \sqrt{\frac{2k \log(2/\delta')}{n}} \stackrel{def}{=} \frac{\sqrt{2kc_n}}{\sqrt{n}}. \tag{15}$$

*Proof.* See [Wei+03]. □

**Lemma 2** (Good Event). *With probability at least $1 - \delta$ the following events holds true for all $(s, a)$ pairs for all episodes of the algorithm:*

$$|\widehat{r}_k(s,a) - r(s,a)| \leq \sqrt{\frac{2c_n \left(\operatorname{Var} R(s,a)\right)}{n_{sa}}} + \frac{c_n}{3(n_{sa} - 1)} \tag{16}$$

$$| \left(\widehat{p}_k(s,a) - p(s,a)\right)^\top V^\star| \leq \sqrt{\frac{2c_n \left(\operatorname{Var}_{p(s,a)} V^\star\right)}{n_{sa}}} + \frac{c_n}{3(1 - \gamma)(n_{sa} - 1)} \tag{17}$$

$$\left| \sqrt{\operatorname*{Var}_{\widehat{p}_k(s,a)} V^\star} - \sqrt{\operatorname*{Var}_{p(s,a)} V^\star} \right| \leq \frac{1}{1 - \gamma} \sqrt{\frac{2c_n}{n - 1}} \tag{18}$$

$$\|\widehat{p}_k(s,a) - p(s,a)\|_1 \leq \frac{\sqrt{2Sc_n}}{\sqrt{n}} \tag{19}$$

*Proof.* Using propositions 1,2,3 and corollary 1 with a union bound over the $(s, a)$ pairs and over the maximum number of samples $n_{sa}$. □

# C Preliminaries

In this section we recall some standard results in reinforcement learning. The results have been adapted so that they could be expressed in terms of a starting distribution $\rho$ instead of a fixed starting state.

## C.1 Sum of Losses

The lemma below expresses the difference in values between two policies as a sum of the per-step losses:

**Lemma 1** (Sum of Losses). *It holds that:*

$$(V^\star - V^\pi)(\rho) = \sum_{(s,a)} \overline{w}_{sa}^{\pi,\rho} (\underbrace{Q^\star(s,\pi^\star(s)) - Q^\star(s,a)}_{\overset{def}{=} \Delta_{sa}}) = \sum_{(s,a)} \overline{w}_{sa}^{\pi,\rho} \Delta_{sa} \tag{20}$$

*Proof.* Consider a fixed starting state $s$. We have that:

$$(V^\star - V^\pi)(s) = r(s,\pi^\star(s)) - r(s,\pi(s)) + \gamma p(s,\pi^\star(s))^\top V^\star - \gamma p(s,\pi(s))^\top V^\pi \tag{21}$$
$$= r(s,\pi^\star(s)) - r(s,\pi(s)) + \gamma(p(s,\pi^\star(s)) - p(s,\pi(s)))^\top V^\star + \tag{22}$$
$$+ \gamma p(s,\pi(s))^\top (V^\star - V^\pi) \tag{23}$$

Induction with a $\rho$-weighted average over the starting state and the definition of $Q^\star$ values (and their gaps $\Delta_{sa}$) conclude the proof. □

## C.2 Simulation Lemmas

The Simulation Lemma below allows to evaluate policy $\pi$ on two different MDPs, with the induced distribution evaluated on the empirical MDP and the value function for the backup on the real MDP.

**Lemma 3** (Simulation Lemma). *It holds that:*

$$(\widehat{V}_k^\pi - V^\pi)(\rho) = \sum_{(s,a)} \widehat{\overline{w}}_{sa}^{\pi,k,\rho} \left( \widehat{r}_k(s,a) - r(s,a) + \gamma(\widehat{p}_k(s,a) - p(s,a))^\top V^\pi \right) \tag{24}$$

*Proof.* From any starting state $s$:

$$(\widehat{V}_k^\pi - V^\pi)(s) = \widehat{r}_k(s,a) - r(s,a) + \gamma(\widehat{p}_k(s,a)^\top \widehat{V}_k^\pi - p(s,a)^\top V^\pi) \tag{25}$$
$$= \widehat{r}_k(s,a) - r(s,a) + \gamma(\widehat{p}_k(s,a) - p(s,a))^\top V^\pi + \gamma \widehat{p}_k(s,a)^\top (\widehat{V}_k^\pi - V^\pi) \tag{26}$$

Induction and a re-weighting by $\rho$ concludes the proof.

□

The following lemma is a consequence of a lemma in [AMK12], and explains that to properly estimate the value function we need to estimate the rewards and transitions accurately only for the optimal policy and the optimal policy on the empirical MDP. Importantly, the lemma uses the true optimal value function $V^\star$.

**Lemma 2** (Simulation Lemma for Optimal Value Function Estimate [GMK13]). *With probability at least $1 - \delta$, outside the failure event for any starting distribution $\rho$ it holds that:*

$$(V^\star - \widehat{V}_k^\star)(\rho) \leq \sum_{(s,a)} \widehat{\overline{w}}_{sa}^{\pi^\star,k,\rho} \left( (r - \widehat{r}_k)(s,a) + \gamma(p - \widehat{p}_k)(s,a)^\top V^\star \right) \leq \sum_{(s,a)} \widehat{\overline{w}}_{sa}^{\pi^\star,k,\rho} CI_{sa}(n_{sa}^k)$$

$$(V^\star - \widehat{V}_k^\star)(\rho) \geq \sum_{(s,a)} \widehat{\overline{w}}_{sa}^{\widehat{\pi}_k^\star,k,\rho} \left( (r - \widehat{r}_k)(s,a) + \gamma(p - \widehat{p}_k)(s,a)^\top V^\star \right) \geq -\sum_{(s,a)} \widehat{\overline{w}}_{sa}^{\widehat{\pi}_k^\star,k,\rho} CI_{sa}(n_{sa}^k)$$

*Proof.* Lemma 2 in [AMK12] gives (here $\widehat{\widetilde{w}}_{sa}^{\pi,k,s_0}$ is the discounted sum of visit probabilities upon starting from $s_0$ and following policy $\pi$ on the empirical MDP $\widehat{\mathcal{M}}_k$):

$$V^\star(s_0) - \widehat{V}_k^\star(s_0) \le \sum_{(s,a)} \widehat{\widetilde{w}}_{sa}^{\pi^\star,k,s_0} \left( \widehat{r}_k(s,a) - r(s,a) + \gamma \left( \widehat{p}_k(s,a) - p(s,a) \right)^\top V^\star \right) \qquad (27)$$

$$V^\star(s_0) - \widehat{V}_k^\star(s_0) \ge \sum_{(s,a)} \widehat{\widetilde{w}}_{sa}^{\widehat{\pi}_k^\star,k,s_0} \left( \widehat{r}_k(s,a) - r(s,a) + \gamma \left( \widehat{p}_k(s,a) - p(s,a) \right)^\top V^\star \right) \qquad (28)$$

$$(29)$$

Outside the failure event (lemma 2) it holds that:

$$V^\star(s_0) - \widehat{V}_k^\star(s_0) \le \sum_{(s,a)} \widehat{\widetilde{w}}_{sa}^{\pi^\star,k,s_0} CI_{sa}(n_{sa}^k) \qquad (30)$$

$$V^\star(s_0) - \widehat{V}_k^\star(s_0) \ge \sum_{(s,a)} \widehat{\widetilde{w}}_{sa}^{\widehat{\pi}_k^\star,k,s_0} CI_{sa}(n_{sa}^k) \qquad (31)$$

Finally, a weighted sum over the probabilities of starting at each starting state $\rho_s$ yields the thesis. $\qquad \square$

Next, we recall the following version of the simulation lemma that expresses the accuracy with which a generic policy $\pi$ can be evaluated on the empirical vs real MDP as a function of its distance to the optimal value function.

**Lemma 4** (Simulation Lemma for Policy Estimate). *If $n_{min}$ is the minimum number of samples allocated to any $(s,a)$ pair then outside of the failure event it holds that:*

$$\|(\widehat{V}_k^\pi - V^\pi)\|_\infty \le \sum_{(s,a)} \widehat{\widetilde{w}}_{sa}^{\pi,k,\rho} CI_{sa}(n_{sa}^k) + Cp(n_{min}) \| (V^\pi - V^\star) \|_\infty. \qquad (32)$$

*Proof.* Using the simulation lemma (lemma 3)

$$(\widehat{V}_k^\pi - V^\pi)(\rho) = \sum_{(s,a)} \widehat{\widetilde{w}}_{sa}^{\pi,k,\rho} \left( \widehat{r}_k(s,a) - r(s,a) + \gamma \left( \widehat{p}_k(s,a) - p(s,a) \right)^\top V^\pi \right) \qquad (33)$$

$$= \sum_{(s,a)} \widehat{\widetilde{w}}_{sa}^{\pi,k,\rho} \left( \widehat{r}_k(s,a) - r(s,a) + \gamma \left( \widehat{p}_k(s,a) - p(s,a) \right)^\top V^\star \right) \qquad (34)$$

$$+ \gamma \sum_{(s,a)} \widehat{\widetilde{w}}_{sa}^{\pi,k,\rho} \left( \widehat{p}_k(s,a) - p(s,a) \right)^\top (V^\pi - V^\star). \qquad (35)$$

Notice that we have the following upper bound outside of the failure event:

$$\left| \sum_{(s,a)} \widehat{\widetilde{w}}_{sa}^{\pi,k,\rho} \left( \widehat{r}_k(s,a) - r(s,a) + \gamma \left( \widehat{p}_k(s,a) - p(s,a) \right)^\top V^\star \right) \right| \le \sum_{(s,a)} \widehat{\widetilde{w}}_{sa}^{\pi,k,\rho} CI_{sa}(n_{sa}^k). \qquad (36)$$

To bound the second line, if $n \ge n_{min}$ we obtain the upper bound (by Holder's inequality):

$$\le \frac{\gamma}{(1-\gamma)} \max_{(s,a)} \| \left( \widehat{p}_k(s,a) - p(s,a) \right) \|_1 \| (V^\pi - V^\star) \|_\infty \overset{def}{=} Cp(n_{min}) \| (V^\pi - V^\star) \|_\infty. \qquad (37)$$

$$\square$$

## C.3 Variance Lemma

The lemma below allows to express how much the variance varies when we consider different value functions.

**Lemma 5.** *For any two random variables $V_1, V_2$ it holds that:*

$$\left| \sqrt{\mathrm{Var}(V_1)} - \sqrt{\mathrm{Var}(V_2)} \right| \le \|V_1 - V_2\|_{2,p} \le \|V_1 - V_2\|_\infty \qquad (38)$$

*where $\| \cdot \|_{2,p}$ denotes the 2-norm of the random variables (i.e., the second moment) under $p$ and $\| \cdot \|_\infty$ is the almost sure upper bound to the random variable.*

*Proof.* Consider the mean-centered random variables $\overline{V}_1 = V_1 - \mathbb{E}\,V_1$ and $\overline{V}_2 = V_2 - \mathbb{E}\,V_2$. Then:

$$\sqrt{\mathrm{Var}(V_1)} = \sqrt{\mathrm{Var}(\overline{V}_1)} = \sqrt{\mathbb{E}(\overline{V}_1)^2} = \|\overline{V}_1\|_{2,p} = \|\overline{V}_2 + \overline{V}_1 - \overline{V}_2\|_{2,p} \tag{39}$$

$$\leq \|\overline{V}_2\|_{2,p} + \|\overline{V}_1 - \overline{V}_2\|_{2,p} = \sqrt{\mathbb{E}(\overline{V}_2)} + \sqrt{\mathbb{E}(\overline{V}_1 - \overline{V}_2)^2} \tag{40}$$

$$= \sqrt{\mathrm{Var}(\overline{V}_2)} + \sqrt{\mathbb{E}(V_1 - V_2)^2 - (\mathbb{E}(V_1 - V_2))^2} \tag{41}$$

$$= \sqrt{\mathrm{Var}(V_2)} + \sqrt{\mathrm{Var}(V_1 - V_2)}. \tag{42}$$

where the inequality is Minkowski's inequality (i.e., the triangle inequality for norm of random variables).

$\square$

# D Optimization Programs

In this section we describe the optimization programs that we investigate in this work: (1) the oracle optimization program, which is directly tied with the accuracy of estimating the optimal value function on the empirical MDP, (2) the algorithm optimization program, which can be solved using the empirical quantities and finally (3) the $\star$-optimization program, function of problem dependent quantities which can be used to analyze the sample complexity of our algorithm.

## D.1 Definitions

**Definition 4** (Oracle Minimax Program).

$$\min_n f_{\mathcal{O}}(n)$$
$$s.t. \quad \sum_{(s,a)} n_{sa} \leq n_{max} \tag{43}$$

*where:*

$$f_{\mathcal{O}}(n) \overset{def}{=} \max_{\rho,\pi} \sum_{(s,a)} \widehat{\overline{w}}_{sa}^{\pi,k+1,\rho} CI_{sa}(n_{sa}^{k+1})$$
$$s.t. \quad (I - \gamma(\widehat{P}_{k+1}^{\pi})^{\top})\widehat{\overline{w}}^{\pi,k+1,\rho} = \rho$$
$$\sum_s \rho_s = 1 \tag{44}$$
$$\rho_{sa} \geq 0$$
$$(\widehat{V}_k^{\star} - \widehat{V}_k^{\pi})(\rho) \leq C\epsilon_k$$

**Definition 5** (Algorithm Minimax Program).

$$\min_n f_{\mathcal{A}}(n)$$
$$s.t. \quad \sum_{(s,a)} n_{sa} \leq n_{max} \tag{45}$$

*where:*

$$f_{\mathcal{A}}(n) \overset{def}{=} \max_{\rho,\pi} \sum_{(s,a)} \widehat{\overline{w}}_{sa}^{\pi,k,\rho}(\widehat{CI}_{sa}(n_{sa}^{k+1}) + B_{ksa}) + 2Cp(n_{min})\epsilon_k^{\pi}$$
$$s.t. \quad (I - \gamma(\widehat{P}_k^{\pi})^{\top})\widehat{\overline{w}}^{\pi,k,\rho} = \rho$$
$$\sum_s \rho_s = 1 \tag{46}$$
$$\rho_{sa} \geq 0$$
$$(\widehat{V}_k^{\star} - \widehat{V}_k^{\pi})(\rho) \leq C\epsilon_k$$

*The program is solved with $\epsilon_k^{\pi} = \epsilon_k$.*

**Definition 6** ($\star$-Minimax Program).

$$\min_n f_{\star}(n)$$
$$s.t. \quad \sum_{(s,a)} n_{sa} \leq n_{max} \tag{47}$$

*where:*

$$f_{\star}(n) \overset{def}{=} \max_{\rho,\pi} \sum_{(s,a)} \overline{w}_{sa}^{\pi,\rho} \left(CI_{sa}(n_{sa}^{k+1}) + 2B_{ksa}\right) + 15Cp(n_{min})\epsilon_k^{\pi} + 8Cp(n_{min})\epsilon_k$$
$$s.t. \quad (I - \gamma(P^{\pi})^{\top})\overline{w}^{\pi,\rho} = \rho$$
$$\sum_s \rho_s = 1 \tag{48}$$
$$\rho_s \geq 0$$
$$(V^{\star} - V^{\pi})(\rho) \leq C^{\star}\epsilon_k$$

*Similarly as above, $\epsilon_k^\pi = \epsilon_k$ when computing the sample complexity because the program of definition 5 is solved with $\epsilon_k^\pi = \epsilon_k$.*

## D.2    Relation Between the Optimization Programs

In this section we investigate the relation between the three optimization programs (oracle, algorithm and $\star$). In particular, we show that we can upper bound the objective function of the inner *maximization* program and enlarge its feasibility set as we move from the oracle to the algorithm and finally to the $\star$ program. This ensures that the outer *minimization* is minimizing a function that is increasingly larger (when moving from the oracle to the algorithm and finally to the $\star$ program), giving an upper bound on its value.

**Lemma 6** (Relation Between the Optimization Programs). *Consider the three optimization programs of definition 4,5,6. We have that:*

$$f_{\mathcal{O}}(n) \leq f_{\mathcal{A}}(n) \tag{49}$$

*Furthermore, outside of the failure event if*

$$|(V^\star - \widehat{V}_k^\star)(\rho)| \leq \epsilon_k, \quad \forall \rho \geq 0, \|\rho\|_1 = 1 \tag{50}$$

*holds then*

$$f_{\mathcal{A}}(n) \leq f_\star(n) \tag{51}$$

$$\tag{52}$$

*also holds.*

*Proof.*

**Oracle Minimax to Algorithm Minimax**    Consider the maximization program contained in the definition of $f_{\mathcal{O}}$, see definition 4. First, we can add the variable $\widehat{\widetilde{w}}^{\pi,k,\rho}$ and the constraint

$$(I - \gamma(\widehat{P}_k^\pi)^\top)\widehat{\widetilde{w}}^{\pi,k,\rho} = \rho \tag{53}$$

to the oracle inner maximization program without changing its objective value or restricting its feasibility set since $\widehat{\widetilde{w}}^{\pi,k,\rho}$ is fully determined by equation 53. Next, lemma 9 allows us to move from using the distribution from episode $k+1$ (which is unknown) to episode $k$ (which can be computed by using the empirical MDP $\widehat{\mathcal{M}}_k$) in the objective function:

$$\sum_{(s,a)} \widehat{\widetilde{w}}_{sa}^{\pi,k+1,\rho} CI_{sa}(n_{sa}^{k+1}) = \sum_{(s,a)} \widehat{\widetilde{w}}_{sa}^{\pi,k,\rho} CI_{sa}(n_{sa}^{k+1}) + \sum_{(s,a)} (\widehat{\widetilde{w}}_{sa}^{\pi,k+1,\rho} - \widehat{\widetilde{w}}_{sa}^{\pi,k,\rho}) CI_{sa}(n_{sa}^{k+1})$$
$$\tag{54}$$

$$\leq \sum_{(s,a)} \widehat{\widetilde{w}}_{sa}^{\pi,k,\rho} CI_{sa}(n_{sa}^{k+1}) + 2Cp(n_{min})\epsilon_k^\pi \tag{55}$$

At this point we can use the variance correction provided by lemma 11 to obtain:

$$\sum_{(s,a)} \widehat{\widetilde{w}}_{sa}^{\pi,k,\rho} CI_{sa}(n_{sa}^{k+1}) + 2Cp(n_{min})\epsilon_k^\pi \leq \sum_{(s,a)} \widehat{\widetilde{w}}_{sa}^{\pi,k,\rho}(\widehat{CI}_{sa}(n_{sa}^{k+1}) + B_{ksa}) + 2Cp(n_{min})\epsilon_k^\pi$$
$$\tag{56}$$

that uses the empirical quantities only. We can now drop the variable $\widehat{\widetilde{w}}^{\pi,k+1,\rho}$ and its constraint

$$(I - \gamma(\widehat{P}_{k+1}^\pi)^\top)\widehat{\widetilde{w}}^{\pi,k+1,\rho} = \rho \tag{57}$$

since the variable $\widehat{\widetilde{w}}^{\pi,k+1,\rho}$ does not appears elsewhere. Notice that this again does not change the feasible set for $(\pi, \rho)$.

**Algorithm Minimax to ⋆-Minimax**   First, we add the variable $\overline{w}^{\pi,\rho}$ and the constraint

$$(I - \gamma(P^\pi)^\top)\overline{w}^{\pi,\rho} = \rho \tag{58}$$

to the minimax optimization program of definition 5; this does not change the objective function or restrict the set of feasible $\rho, \pi$ since equation 58 can always be satisfied: $\overline{w}^{\pi,\rho}$ is the distribution induced by the policy upon starting from $\rho$ on $\mathcal{M}$. Next, we chain lemma 9 with lemma 11 to express the objective function as a function of the "real quantities", obtaining:

$$\sum_{(s,a)} \widehat{\overline{w}}_{sa}^{\pi,k,\rho} \widehat{CI}_{sa}(n_{sa}^{k+1}) \leq \sum_{(s,a)} \overline{w}_{sa}^{\pi,\rho}\left(CI_{sa}(n_{sa}^{k+1}) + B_{ksa}\right) + Cp(n_{min})\epsilon_k^\pi. \tag{59}$$

Now we examine the remaining term:

$$\sum_{(s,a)} \left(\overline{w}_{sa}^{\pi,\rho} - \widehat{\overline{w}}_{sa}^{\pi,k,\rho}\right)(2B_{ksa}) = 2\sum_{(s,a)} \left(\overline{w}_{sa}^{\pi,\rho} - \widehat{\overline{w}}_{sa}^{\pi,k,\rho}\right)\left(\frac{2c_n}{(1-\gamma)(n_{sa}-1)} + \gamma\sqrt{\frac{2c_n}{n_{sa}}}\epsilon_k\right). \tag{60}$$

We further split the above into two. For the first, using the definition of $CI_{sa}(n_{sa}^k)$ and lemma 10 we obtain:

$$2\sum_{(s,a)} \left(\overline{w}_{sa}^{\pi,\rho} - \widehat{\overline{w}}_{sa}^{\pi,k,\rho}\right)\left(\frac{2c_n}{(1-\gamma)(n_{sa}-1)}\right) = 2\sum_{(s,a)} \left(\overline{w}_{sa}^{\pi,\rho} - \widehat{\overline{w}}_{sa}^{\pi,k,\rho}\right)\left(6 \times \underbrace{\frac{c_n}{3(1-\gamma)(n_{sa}-1)}}_{CI_{sa}(n_{sa}^k)}\right) \tag{61}$$

$$\leq 12\sum_{(s,a)} \left(\overline{w}_{sa}^{\pi,\rho} - \widehat{\overline{w}}_{sa}^{\pi,k,\rho}\right)CI_{sa}(n_{sa}^k) \leq 12Cp(n_{min})\epsilon_k^\pi. \tag{62}$$

For the second term, use the definition of $Cp(n_{min})$ together with $\sum_{(s,a)} \left(\overline{w}_{sa}^{\pi,\rho} + \widehat{\overline{w}}_{sa}^{\pi,k,\rho}\right) = \frac{2}{(1-\gamma)}$ to claim:

$$2\sum_{(s,a)} \left(\overline{w}_{sa}^{\pi,\rho} - \widehat{\overline{w}}_{sa}^{\pi,k,\rho}\right)\left(\gamma\sqrt{\frac{2c_n}{n_{sa}}}\epsilon_k\right) \leq 4\sum_{(s,a)} \left(\overline{w}_{sa}^{\pi,\rho} + \widehat{\overline{w}}_{sa}^{\pi,k,\rho}\right)(1-\gamma)Cp(n_{min})\epsilon_k = 8Cp(n_{min})\epsilon_k. \tag{63}$$

In summary we have obtained:

$$\sum_{(s,a)} \widehat{\overline{w}}_{sa}^{\pi,k,\rho}\left(\widehat{CI}_{sa}(n_{sa}^{k+1}) + B_{ksa}\right) + 2Cp(n_{min}) \tag{64}$$

$$\leq \sum_{(s,a)} \overline{w}_{sa}^{\pi,\rho}\left(CI_{sa}(n_{sa}^{k+1}) + 2B_{ksa}\right) + 15Cp(n_{min})\epsilon_k^\pi + 8Cp(n_{min})\epsilon_k. \tag{65}$$

At this point we can drop the the variable $\widehat{\overline{w}}^{\pi,k,\rho}$ and its constraint:

$$(I - \gamma(\widehat{P}_{k+1}^\pi)^\top)\widehat{\overline{w}}^{\pi,k,\rho} = \rho \tag{66}$$

since the variable $\widehat{\overline{w}}^{\pi,k,\rho}$ does not show up elsewhere. Notice that this operation does not change the constraints on the $\rho, \pi$ optimization variables. Finally, lemma 7 allows us to replace the constraint $(\widehat{V}_k^\star - \widehat{V}_k^\pi)(\rho) \leq C\epsilon_k$ with the relaxed version on the real MDP $(V^\star - V^\pi)(\rho) \leq C^\star\epsilon_k$. This enlarges the feasibility set for the policies. Notice that an enlarged feasibility for the maximization program set can only increase the objective function, so $f_\mathcal{O} \leq f_\star$ holds pointwise.   $\square$

# E  Helper Lemmas

In this section we state and prove some helper lemmas.

## E.1  Enlarging The Feasibility Set

**Lemma 7** (Enlarging The Feasibility Set). *Outside of the failure event if:*

$$|(V^\star - \widehat{V}_k^\star)(\rho)| \leq \epsilon_k, \quad \forall \rho \geq 0, \|\rho\|_1 = 1 \tag{67}$$

$$\pi \in \widehat{S}_k \tag{68}$$

$$\epsilon_k^\pi \leq \epsilon_k \tag{69}$$

*then*

$$\pi \in S_k. \tag{70}$$

*Proof.* Since by assumption:

$$(\widehat{V}_k^\star - \widehat{V}_k^\pi)(\rho) \leq C\epsilon_k \tag{71}$$

for all $\rho$ then must we have that (by choosing $\rho$ to be any canonical vector in $\mathbb{R}^{|\mathcal{S}|}$):

$$\|\widehat{V}_k^\star - \widehat{V}_k^\pi\|_\infty \leq C\epsilon_k. \tag{72}$$

Next, for an arbitrary $\rho$ consider:

$$(\widehat{V}_k^\star - \widehat{V}_k^\pi)(\rho) = (\underbrace{\widehat{V}_k^\star - V^\star}_{\geq -\epsilon_k} + V^\star - V^\pi + V^\pi - \widehat{V}_k^\pi)(\rho) \tag{73}$$

and apply the simulation lemma, lemma 4 to the last difference obtaining:

$$\geq -\epsilon_k + (V^\star - V^\pi)(\rho) - \epsilon_k - Cp(n_{min})\|V^\star - V^\pi\|_\infty \tag{74}$$

Therefore:

$$-2\epsilon_k + (V^\star - V^\pi)(\rho) - Cp(n_{min})\|V^\star - V^\pi\|_\infty \leq (\widehat{V}_k^\star - \widehat{V}_k^\pi)(\rho) \leq C\epsilon_k \tag{75}$$

from which we can derive:

$$(V^\star - V^\pi)(\rho) \leq C\epsilon_k + 2\epsilon_k + Cp(n_{min})\|V^\star - V^\pi\|_\infty. \tag{76}$$

By taking $\rho$ to be each canonical vector in $\mathbb{R}^{\mathcal{S}}$ we can derive:

$$\|V^\star - V^\pi\|_\infty \leq C\epsilon_k + 2\epsilon_k + Cp(n_{min})\|V^\star - V^\pi\|_\infty. \tag{77}$$

from which

$$\|V^\star - V^\pi\|_\infty \leq \frac{C\epsilon_k + 2\epsilon_k}{1 - Cp(n_{min})}. \tag{78}$$

follows.[8] Since this is a max-norm bound on the vector $V^\star - V^\pi$, a linear combination weighted by $\rho$ must also satisfy:

$$(V^\star - V^\pi)(\rho) \leq \frac{C + 2}{1 - Cp(n_{min})}\epsilon_k \overset{def}{=} C^\star \epsilon_k. \tag{79}$$

The thesis finally follows by definition of $S_k$. □

## E.2 Visit Probability Lemma

Although the visit distribution $\overline{w}^{\pi,\rho}$ can require many samples to be estimated accurately, in this section we show that the uncertainty in the distribution nicely interacts with the confidence intervals for the transitions and rewards. We need the following helper lemma:

**Lemma 8** (Distribution Lemma). *It holds that:*

$$\left(\overline{w}^{\pi,\rho} - \widehat{\overline{w}}^{\pi,k,\rho}\right)^{\top} = (\overline{w}^{\pi,\rho})^{\top}(P^{\pi} - \widehat{P}_k^{\pi})\sum_{t=0}^{\infty}\gamma^{t+1}\left(\widehat{P}_k^{\pi}\right)^t. \tag{80}$$

*Proof.* The cumulative discounted sum of visit probabilities (for policy $\pi$ upon starting from $\rho$) satisfies (see for example [WBS07])

$$\overline{w}^{\pi,\rho} = \rho + \gamma(P^{\pi})^{\top}\overline{w}^{\pi,\rho} \tag{81}$$

$$\widehat{\overline{w}}^{\pi,k,\rho} = \rho + \gamma(\widehat{P}^{\pi})^{\top}\widehat{\overline{w}}^{\pi,k,\rho} \tag{82}$$

on $\mathcal{M}$ and $\widehat{\mathcal{M}}_k$, respectively. Subtraction yields:

$$\overline{w}^{\pi,\rho} - \widehat{\overline{w}}^{\pi,k,\rho} = \gamma\left((P^{\pi})^{\top}\overline{w}^{\pi,\rho} - (\widehat{P}_k^{\pi})^{\top}\widehat{\overline{w}}^{\pi,k,\rho}\right) \tag{83}$$

$$= \gamma\left((P^{\pi})^{\top}\overline{w}^{\pi,\rho} - (\widehat{P}_k^{\pi})^{\top}\overline{w}^{\pi,\rho} + (\widehat{P}_k^{\pi})^{\top}\overline{w}^{\pi,\rho} - (\widehat{P}_k^{\pi})^{\top}\widehat{\overline{w}}^{\pi,k,\rho}\right) \tag{84}$$

$$= \gamma\left((P^{\pi} - \widehat{P}_k^{\pi})^{\top}\overline{w}^{\pi,\rho} + (\widehat{P}_k^{\pi})^{\top}\left(\overline{w}^{\pi,\rho} - \widehat{\overline{w}}^{\pi,k,\rho}\right)\right). \tag{85}$$

By induction, this yields:

$$\overline{w}^{\pi,\rho} - \widehat{\overline{w}}^{\pi,k,\rho} = \sum_{t=0}^{\infty}\gamma^{t+1}\left((\widehat{P}_k^{\pi})^{\top}\right)^t\left((P^{\pi} - \widehat{P}_k^{\pi})^{\top}\overline{w}^{\pi,\rho}\right). \tag{86}$$

By transposing the above equality we obtain the statement. $\square$

Now we are ready to analyze how the distribution shift interacts with the confidence intervals for policies that are accurately estimated:

**Lemma 9** (Interaction between distribution inaccuracy and confidence intervals). *If for $\pi$ it holds that*

$$(\widehat{\overline{w}}^{\pi,k,\rho})^{\top}CI^k \leq \epsilon_k^{\pi} \tag{87}$$

*then we have that:*

$$\left(\overline{w}^{\pi,\rho} - \widehat{\overline{w}}^{\pi,k,\rho}\right)^{\top}CI^k \leq Cp(n_{min})\epsilon_k^{\pi}$$

$$\left(\widehat{\overline{w}}^{\pi,j,\rho} - \widehat{\overline{w}}^{\pi,k,\rho}\right)^{\top}CI^k \leq 2Cp(n_{min})\epsilon_k^{\pi}, \quad j \geq k. \tag{88}$$

*Proof.* Thanks to lemma 8 we can write:

$$\left(\overline{w}^{\pi,\rho} - \widehat{\overline{w}}^{\pi,k,\rho}\right)^{\top}CI^k = \gamma\,(\overline{w}^{\pi,\rho})^{\top}(P^{\pi} - \widehat{P}_k^{\pi})\sum_{t=0}^{\infty}\gamma^t\left(\widehat{P}_k^{\pi}\right)^t CI^k \tag{89}$$

$$\leq \gamma\|(\overline{w}^{\pi,\rho})^{\top}(P^{\pi} - \widehat{P}_k^{\pi})\|_1\|\sum_{t=0}^{\infty}\gamma^t\left(\widehat{P}_k^{\pi}\right)^t CI^k\|_{\infty} \tag{90}$$

Notice that the $j$-th row of

$$\sum_{t=0}^{\infty}\gamma^t\left(\widehat{P}_k^{\pi}\right)^t \tag{91}$$

is precisely the discounted sum of visit probabilities upon starting from state $j$ on $\widehat{\mathcal{M}}_k$ and following $\pi$. Let us call $e_j$ the canonical vector with a 1 in position $j$ and 0's elsewhere; the $j$-th row is expressible as:

$$e_j^\top \sum_{t=0}^\infty \gamma^t \left(\widehat{P}_k^\pi\right)^t = (\widehat{\overline{w}}^{\pi,k,s_j})^\top. \tag{92}$$

This immediately yields:

$$e_j^\top \sum_{t=0}^\infty \gamma^t \left(\widehat{P}_k^\pi\right)^t CI^k = (\widehat{\overline{w}}^{\pi,k,s_j})^\top CI^k \le \epsilon_k^\pi. \tag{93}$$

and therefore equation 89 admits the upper bound (by Holder's inequality):

$$\le \gamma \|(\overline{w}^{\pi,\rho})^\top (P^\pi - \widehat{P}_k^\pi)\|_1 \epsilon_k^\pi \tag{94}$$

$$= \gamma \| \sum_{(s,a)} \overline{w}_{sa}^{\pi,\rho} \left(p(s,a) - \widehat{p}_k(s,a)\right)^\top \|_1 \epsilon_k^\pi \tag{95}$$

$$\le \gamma \sum_{(s,a)} \overline{w}_{sa}^{\pi,\rho} \| \left(p(s,a) - \widehat{p}_k(s,a)\right)^\top \|_1 \epsilon_k^\pi \tag{96}$$

$$\le Cp(n_{min})\epsilon_k^\pi. \tag{97}$$

To obtain the second equation of 88 proceed similarly:

$$\left(\widehat{\overline{w}}^{\pi,j,\rho} - \widehat{\overline{w}}^{\pi,k,\rho}\right)^\top CI^k = \gamma \left(\widehat{\overline{w}}^{\pi,j,\rho}\right)^\top (\widehat{P}_j^\pi - \widehat{P}_k^\pi) \sum_{t=0}^\infty \gamma^t \left(\widehat{P}_k^\pi\right)^t CI^k \tag{98}$$

$$\le \gamma \| \left(\widehat{\overline{w}}^{\pi,j,\rho}\right)^\top (\widehat{P}_j^\pi - \widehat{P}_k^\pi)\|_1 \| \sum_{t=0}^\infty \gamma^t \left(\widehat{P}_k^\pi\right)^t CI^k\|_\infty \tag{99}$$

$$\le \gamma \sum_{(s,a)} \widehat{\overline{w}}_{sa}^{\pi,j,\rho} \left(\|\widehat{p}_j(s,a) - p(s,a)\|_1 + \|p(s,a) - \widehat{p}_k(s,a)\|_1\right) \| \sum_{t=0}^\infty \gamma^t \left(\widehat{P}_k^\pi\right)^t CI^k\|_\infty$$
$$\tag{100}$$

$$\le 2Cp(n_{min})\epsilon_k^\pi. \tag{101}$$

$\square$

### E.3 Bernstein Correction

In this section we discuss how to correct the variance estimate computed with the empirical transitions and value function estimate in a way that results in a variance overestimate (to obtain valid confidence intervals).

**Lemma 10** (Bernstein Correction). *If*

$$\|V^\star - \widehat{V}_k^\star\|_\infty \le \epsilon_k \tag{102}$$

*holds then outside of the failure event we have that:*

$$\left| \sqrt{\mathop{\mathrm{Var}}_{p(s,a)} V^\star} - \sqrt{\mathop{\mathrm{Var}}_{\widehat{p}_k(s,a)} \widehat{V}_k^\star} \right| \le \epsilon_k + \frac{1}{1-\gamma} \sqrt{\frac{2c_n}{n_{sa} - 1}} \tag{103}$$

*Proof.* Outside the failure event:

$$\left| \sqrt{\mathop{\mathrm{Var}}_{\widehat{p}_k(s,a)} V^\star} - \sqrt{\mathop{\mathrm{Var}}_{p(s,a)} V^\star} \right| \le \frac{1}{1-\gamma} \sqrt{\frac{2c_n}{n_{sa} - 1}} \tag{104}$$

holds and further lemma 5 yields:

$$\left| \sqrt{\mathop{\mathrm{Var}}_{\widehat{p}_k(s,a)} V^\star} - \sqrt{\mathop{\mathrm{Var}}_{\widehat{p}_k(s,a)} \widehat{V}_k^\star} \right| \le \|V^\star - \widehat{V}_k^\star\|_\infty. \tag{105}$$

Chaining the two yields the statement. $\square$

**Lemma 11** (Realation Between Real and Empirical Confidence Intervals)**.** *If*

$$\|V^\star - \widehat{V}_k^\star\|_\infty \le \epsilon_k \tag{106}$$

*holds then outside of the failure event we have that:*

$$CI_{sa}(n_{sa}) \le \widehat{CI}_{sa}(n_{sa}) + B_{ksa} \le CI_{sa}(n_{sa}) + 2B_{ksa} \tag{107}$$

*where:*

$$B_{ksa} \overset{def}{=} \frac{2c_n}{(1-\gamma)(n_{sa}-1)} + \gamma\sqrt{\frac{2c_n}{n_{sa}}}\epsilon_k. \tag{108}$$

*Proof.* Apply lemma 10 twice, first to obtain the "hat quantities" and then to go back to the "real quantities":

$$CI_{sa}(n_{sa}) \overset{def}{=} \sqrt{\frac{2\operatorname{Var}R(s,a)c_n}{n_{sa}}} + \gamma\sqrt{\frac{2\operatorname{Var}_{p(s,a)}V^\star c_n}{n_{sa}}} + \underbrace{\frac{c_n}{3(n_{sa}-1)} + \frac{\gamma c_n}{3(1-\gamma)(n_{sa}-1)}}_{=\frac{c_n}{3(1-\gamma)(n_{sa}-1)}} \tag{109}$$

$$\le \sqrt{\frac{2\operatorname{Var}\widehat{R}(s,a)c_n}{n_{sa}}} + \gamma\sqrt{\frac{2\operatorname{Var}_{\widehat{p}_k(s,a)}\widehat{V}_k^\star c_n}{n_{sa}}} + \frac{c_n}{3(1-\gamma)(n_{sa}-1)} + \underbrace{\frac{2c_n}{(1-\gamma)(n_{sa}-1)} + \gamma\sqrt{\frac{2c_n}{n_{sa}}}\epsilon_k}_{B_{ksa}} \tag{110}$$

$$\overset{def}{=} \widehat{CI}_{sa}(n_{sa}) + B_{ksa} \tag{111}$$

$$\le \sqrt{\frac{2\operatorname{Var}R(s,a)c_n}{n_{sa}}} + \gamma\sqrt{\frac{2\operatorname{Var}_{p(s,a)}V^\star c_n}{n_{sa}}} + \frac{c_n}{3(1-\gamma)(n_{sa}-1)} + \underbrace{2\left(\frac{2c_n}{(1-\gamma)(n_{sa}-1)} + \gamma\sqrt{\frac{2c_n}{n_{sa}}}\epsilon_k\right)}_{2B_{ksa}} \tag{112}$$

$$\overset{def}{=} CI_{sa}(n_{sa}) + 2B_{ksa}. \tag{113}$$

$\square$

## E.4  Feasible Set Contains Good Policies

In this section we build the supporting lemmas to show that the feasibility set $(\widehat{V}_k^\star - \widehat{V}_k^\pi)(\rho) \le C\epsilon_k$ in episode $k$ is constructed in a way that ensures that the optimal policy $\pi^\star$ and the next-episode empirical optimal policy $\widehat{\pi}_{k+1}^\star$ are never eliminated at step $k$, i.e., they are $C\epsilon_k$-optimal for all starting distribution $\rho$. This ensures that enough samples are allocated at step $k$ to use lemma 2 at the next episode. This guarantees an accurate estimate of the value function.

First we focus on the optimal policy $\pi^\star$.

**Lemma 12** ($\pi^\star$ is a Feasible Solution)**.** *Outside of the failure event, if*

$$\|\widehat{V}_k^\star - V^\star\|_\infty \le \epsilon_k \tag{114}$$

$$\epsilon_k^{\pi^\star} \le \epsilon_k \tag{115}$$

*holds at step $k$ then it holds that:*

$$(\widehat{V}_k^\star - \widehat{V}_k^{\pi^\star})(\rho) \le 2\epsilon_k \le C\epsilon_k, \quad \forall\rho \ge 0, \|\rho\|_1 = 1 \tag{116}$$

*Proof.* Using the simulation lemma (lemma 3) with $\pi = \pi^\star$ and using the fact that we are outside of the failure event we obtain that:

$$|(\widehat{V}_k^\star - V^\star + V^\star - \widehat{V}_k^{\pi^\star})(\rho)| \le |(\widehat{V}_k^\star - V^\star)(\rho)| + |(V^\star - \widehat{V}_k^{\pi^\star})(\rho)| \tag{117}$$

$$\le \epsilon_k + \left|\sum_{(s,a)} \widehat{\overline{w}}_{sa}^{\pi^\star,k,\rho} CI_{sa}(n_{sa}^k)\right| \le \epsilon_k + \epsilon_k^{\pi^\star} \le 2\epsilon_k. \tag{118}$$

$\square$

Next we turn our attention to the next-step empirical optimal policy. To ensure accuracy, we need to show that we always allocate enough samples to $\widehat{\pi}_{k+1}^\star$ at step $k$, i.e., $\widehat{\pi}_{k+1}^\star$ is feasible at step $k$. We achieve this through an inductive argument in theorem 1, which leverages the following lemma. The lemma shows that if a policy is ruled out then it can never become optimal again. This lemma plays a key role in constructing the constraint $(\widehat{V}_k^\star - \widehat{V}_k^\pi)(\rho) \le C\epsilon_k$ because it defines its size through the constant $C$.

**Lemma 13** (Ruled-Out Policies Can Never Be Optimal Again). *Let $j$ be the first episode in which policy $\mu$ is not feasible for some $\rho$ in the sense that $\mu \notin \widehat{S}_j$ while $\mu \in \widehat{S}_{j-1}, \mu \in \widehat{S}_{j-2}, \dots, \mu \in \widehat{S}_1$ holds. Outside of the failure event if $\pi^\star \in \widehat{S}_j, C \ge 20, \epsilon_j = \epsilon_{j-1}/2$ and:*

$$|(V^\star - \widehat{V}_j^\star)(\rho)| \le \epsilon_j, \quad \forall \rho \ge 0, \|\rho\|_1 = 1 \tag{119}$$

$$|(V^\star - \widehat{V}_{j-1}^\star)(\rho)| \le \epsilon_{j-1}, \quad \forall \rho \ge 0, \|\rho\|_1 = 1 \tag{120}$$

*hold together with:*

$$2\left(\epsilon_{j-1}^\mu + 2Cp(n_{min})\epsilon_{j-1}^\mu + Cp(n_{min})(1+C)\epsilon_{j-1}\right) \le 4\epsilon_{j-1} \tag{121}$$

$$2\left(\epsilon_j^{\widehat{\pi}_j^\star} + 2Cp(n_{min})\epsilon_j^{\widehat{\pi}_j^\star} + Cp(n_{min})(1+C)\epsilon_j\right) \le 4\epsilon_j \tag{122}$$

*then $\mu$ cannot be an optimal policy on any empirical MDP $\widehat{\mathcal{M}}_k$ for $k \ge j$.*

*Proof.* Coupled with the hypothesis, Lemma 14 ensures:

$$|(\widehat{V}_k^{\widehat{\pi}_j^\star} - \widehat{V}_j^{\widehat{\pi}_j^\star})(\rho)| \le 4\epsilon_j \tag{123}$$

$$|(\widehat{V}_k^\mu - \widehat{V}_{j-1}^\mu)(\rho)| \le 4\epsilon_{j-1} \tag{124}$$

$$|(\widehat{V}_j^\mu - \widehat{V}_{j-1}^\mu)(\rho)| \le 4\epsilon_{j-1} \tag{125}$$

Since $\mu \notin \widehat{S}_j$ by assumption, we have that for at least a starting distribution $\rho$:

$$(\widehat{V}_j^{\widehat{\pi}_j^\star} - \widehat{V}_j^\mu)(\rho) > C\epsilon_j = \frac{1}{2}C\epsilon_{j-1}. \tag{126}$$

This implies that for that starting distribution $\rho$:

$$(\widehat{V}_k^\mu - \widehat{V}_k^\star)(\rho) \le (\widehat{V}_k^\mu - \widehat{V}_k^{\widehat{\pi}_j^\star})(\rho) \tag{127}$$

$$= (\widehat{V}_k^\mu - \widehat{V}_{j-1}^\mu + \widehat{V}_{j-1}^\mu - \widehat{V}_j^\mu + \widehat{V}_j^\mu - \widehat{V}_j^{\widehat{\pi}_j^\star} + \widehat{V}_j^{\widehat{\pi}_j^\star} - \widehat{V}_k^{\widehat{\pi}_j^\star})(\rho) \tag{128}$$

$$< 8\epsilon_{j-1} + 4\epsilon_j - \frac{1}{2}C\epsilon_{j-1} = 8\epsilon_{j-1} + 2\epsilon_{j-1} - \frac{1}{2}C\epsilon_{j-1} \le 0 \tag{129}$$

In other words $\mu$ is not optimal on $\widehat{\mathcal{M}}_k$. □

The following helper lemma explains that the empirical value of a policy doesn't change a lot between different episodes provided that the policy is feasible for the smaller-numbered episode. In other words, if a policy is accurately estimated, say of order $\epsilon$, its value on all empirical MDPs for later episodes has a fluctuation of order $\epsilon$.

**Lemma 14** (Empirical Value of Feasible Policies Does Not Change Much In Later Episodes). *If $\pi \in \widehat{S}_j$ and*

$$|(V^\star - \widehat{V}_j^\star)(\rho)| \le \epsilon_j, \quad \forall \rho \ge 0, \|\rho\|_1 = 1 \tag{130}$$

*also holds, then outside of the failure event it holds that:*

$$\left|\left(\widehat{V}_k^\pi - \widehat{V}_j^\pi\right)(\rho)\right| \le (2 + 4Cp(n_{min}))\epsilon_j^\pi + 2Cp(n_{min})(1+C)\epsilon_j \tag{131}$$

*for all episodes $k \ge j$.*

*Proof.* The simulation lemma 3 applied to the two empirical MDPs $\widehat{\mathcal{M}}_k$ and $\widehat{\mathcal{M}}_j$ gives:

$$\left(\widehat{V}_k^\pi - \widehat{V}_j^\pi\right)(\rho) = \sum_{(s,a)} \widehat{\overline{w}}_{sa}^{\pi,k,\rho} \left((\widehat{r}_k(s,a) - \widehat{r}_j(s,a)) + \gamma\left(\widehat{p}_k(s,a) - \widehat{p}_j(s,a)\right)^\top \widehat{V}_j^\pi\right) \quad (132)$$

$$= \sum_{(s,a)} \widehat{\overline{w}}_{sa}^{\pi,k,\rho} \left((\widehat{r}_k(s,a) - \widehat{r}_j(s,a)) + \gamma\left(\widehat{p}_k(s,a) - \widehat{p}_j(s,a)\right)^\top V^\star\right) \quad (133)$$

$$+ \gamma \sum_{(s,a)} \widehat{\overline{w}}_{sa}^{\pi,k,\rho} \left(\widehat{p}_k(s,a) - \widehat{p}_j(s,a)\right)^\top \left(\widehat{V}_j^\pi - V^\star\right) \quad (134)$$

We focus on the first term. Outside the failure event the first upper bound below holds

$$\leq \sum_{(s,a)} \widehat{\overline{w}}_{sa}^{\pi,k,\rho} \left(CI_{sa}(n_{sa}^k) + CI_{sa}(n_{sa}^j)\right) \leq 2 \sum_{(s,a)} \widehat{\overline{w}}_{sa}^{\pi,k,\rho} CI_{sa}(n_{sa}^j). \quad (135)$$

The second upper bound above holds because $n_{sa}^k \geq n_{sa}^j$ (i.e., number of samples can only increase) and the confidence intervals are shrinking with increasing samples: $CI_{sa}(n_{sa}^k) \leq CI_{sa}(n_{sa}^j)$. Lemma 9 allows[9] us to use the empirical distribution from step $j$ instead of $k$ ensuring:

$$2 \sum_{(s,a)} \widehat{\overline{w}}_{sa}^{\pi,k,\rho} CI_{sa}(n_{sa}^j) \leq 2 \sum_{(s,a)} \widehat{\overline{w}}_{sa}^{\pi,j,\rho} CI_{sa}(n_{sa}^j) + 4Cp(n_{min})\epsilon_j^\pi \quad (136)$$

$$\leq (2 + 4Cp(n_{min}))\epsilon_j^\pi. \quad (137)$$

The second step is by definition of $\epsilon_j^\pi$. Now we bound the remaining term; by the hypothesis of this lemma:

$$|(\widehat{V}_j^\pi - V^\star)(\rho)| = |(\widehat{V}_j^\pi - \widehat{V}_j^\star + \widehat{V}_j^\star - V^\star)(\rho)| \leq (C+1)\epsilon_j \quad (138)$$

for all $\rho$ and so in particular we must have $\|\widehat{V}_j^\pi - V^\star\|_\infty \leq (1+C)\epsilon_j$ and hence outside of the failure event by Holder (and by adding and subtracting $p(s,a)$) it holds that:

$$\gamma \sum_{(s,a)} \widehat{\overline{w}}_{sa}^{\pi,k,\rho} \left(\widehat{p}_k(s,a) - \widehat{p}_j(s,a)\right)^\top \left(\widehat{V}_j^\star - V^\star\right) \leq 2Cp(n_{min})(1+C)\epsilon_j \quad (139)$$

giving the thesis. $\square$

### E.5 Feasible Policies Will Have Improved Accuracy

In this section we show that feasible policies for the algorithm minimax program of definition 5 will gain accuracy at the next episode provided that they are already accurately estimated in the current episode (condition $\epsilon_k^\pi \leq \epsilon_k$, which is equivalent to a small distribution shift).

**Lemma 15** (Feasible Policies Will Have Improved Accuracy). *Outside of the failure event if:*

$$\pi \in \widehat{S}_k; \quad \epsilon_k^\pi \leq \epsilon_k \quad (140)$$

*holds then at the next episode it holds that:*

$$\epsilon_{k+1}^\pi \leq \epsilon_{k+1} = \frac{\epsilon_k}{2}. \quad (141)$$

*Proof.* BESPOKE solves a program equivalent to the minimax program of definition 5 that ensures that for feasible $(\pi, \rho)$:

$$\frac{\epsilon_k}{4} \leq \sum_{(s,a)} \widehat{\overline{w}}_{sa}^{\pi,k,\rho} \left(\widehat{CI}_{sa}(n_{sa}^k) + B_{ksa}\right) + 2Cp(n_{min})\epsilon_k \leq \frac{\epsilon_k}{2}. \quad (142)$$

by the choice of $n_{max}$ (see inner loop over $n_{max}$ in the algorithm 1). Since $\epsilon_k^\pi \leq \epsilon_k$, we have that:

$$\frac{\epsilon_k}{4} \leq \sum_{(s,a)} \widehat{\overline{w}}_{sa}^{\pi,k,\rho} \left(\widehat{CI}_{sa}(n_{sa}^k) + B_{ksa}\right) + 2Cp(n_{min})\epsilon_k^\pi \leq \frac{\epsilon_k}{2} \quad (143)$$

must hold. By lemma 6 we have that the value of the oracle objective of definition 4 is less than $\epsilon_k/2$, i.e.:

$$\epsilon_{k+1}^{\pi} \overset{def}{=} \sum_{(s,a)} \widehat{w}_{sa}^{\pi,k+1,\rho} CI_{sa}(n_{sa}^{k+1}) \leq \frac{\epsilon_k}{2} \overset{def}{=} \epsilon_{k+1} \tag{144}$$

must hold for those $(\pi, \rho)$. $\qquad\square$

**Lemma 16** (Minimum Number of Samples). *Outside of the failure event if $C = 20$ and*

$$n_{sa} \geq n_{min} \overset{def}{=} \frac{2 \times 625^2 \gamma^2 S c_n}{(1-\gamma)^2} \tag{145}$$

*then it holds that:*

$$Cp(n_{min}) \leq \frac{1}{625} \tag{146}$$

$$2Cp(n_{min})(2+C) \leq \frac{1}{2} \tag{147}$$

*Proof.* Immediate, by definition of $Cp(n_{min})$ and $n_{min}$, see appendix A. $\qquad\square$

Finally, the following technical lemma ensures that if the optimal policy and empirical optimal policy of step $k$ are feasible in all episodes up to $k$ then the accuracy in value function estimate can be guaranteed.

**Lemma 17** (Guaranteed Accuracy). *Outside of the failure event if:*

1. $\pi^\star \in \widehat{S}_j$ *in all episodes $j < k$*

2. $\widehat{\pi}_k^\star \in \widehat{S}_j$ *in all episodes $j < k$*

*then*

$$|(V^\star - \widehat{V}_k^\star)(\rho)| \leq \epsilon_k, \quad \forall \rho \geq 0, \|\rho\|_1 = 1 \tag{148}$$

*holds.*

*Proof.* Chaining lemma 15 for $\pi^\star$ and all episodes up to $k-1$ gives:

$$\max_\rho \sum_{(s,a)} \widehat{w}_{sa}^{\pi^\star,k,\rho} CI_{sa}(n_{sa}^k) \overset{def}{=} \epsilon_k^{\pi^\star} \leq \epsilon_k. \tag{149}$$

Likewise, chaining lemma 15 for $\widehat{\pi}_k^\star$ and all episodes up to $k-1$ gives:

$$\max_\rho \sum_{(s,a)} \widehat{w}_{sa}^{\widehat{\pi}_k^\star,k,\rho} CI_{sa}(n_{sa}^k) \overset{def}{=} \epsilon_k^{\widehat{\pi}_k^\star} \leq \epsilon_k. \tag{150}$$

Finally, lemma 2 gives the thesis. $\qquad\square$

# F  Main Results

In this section we show that the algorithm works as intended, namely it terminates in finite time after logarithmically many iterations and returns with high probability an $\epsilon_{Input}$-correct value function estimate and an almost $\epsilon_{Input}$-suboptimal policy. Finally, we analyze its sample complexity.

## F.1  BESPOKE Works as Intended

**Theorem 1** (BESPOKE Works as Intended). *With probability at least $1 - \delta$, in every episode $k$* BESPOKE *maintains an empirical MDP $\widehat{\mathcal{M}}_k$ such that its optimal value function $\widehat{V}_k^\star$ and its optimal policy $\widehat{\pi}_k^\star$ satisfy:*

$$\|V^\star - \widehat{V}_k^\star\|_\infty \le \epsilon_k, \quad \|V^\star - V^{\widehat{\pi}_k^\star}\|_\infty \le 2\epsilon_k$$

*where $\epsilon_{k+1} \stackrel{def}{=} \frac{\epsilon_k}{2}$, $\forall k$. In particular, when* BESPOKE *terminates in episode $k_{Final}$ it holds that $\frac{\epsilon_{Input}}{2} \le \epsilon_{k_{Final}} \le \epsilon_{Input}$.*

*Proof.* We reason by induction outside of the failure event, which has measure $1 - \delta$ by lemma 2. The inductive hypothesis is over the episodes $k = 1, 2, \dots$ and consists of the following two conditions:

1. $\pi^\star \in \widehat{S}_j$ in all episodes $j < k$

2. $\widehat{\pi}_k^\star \in \widehat{S}_j$ in all episodes $j < k$

In other words, we assume that the optimal policy $\pi^\star$ is feasible up to episode $k - 1$, and that the optimal policy on $\widehat{\mathcal{M}}_k$ is also feasible up to episode $k - 1$. Before showing the inductive step, notice that the inductive hypothesis together with lemma 17 ensures[10]:

$$\|V^\star - \widehat{V}_p^\star\|_\infty \le \epsilon_p, \quad \forall p < k. \tag{151}$$

Notice that the above equation is equivalent to:

$$|(V^\star - \widehat{V}_p^\star)(\rho)| \le \epsilon_p, \quad \forall \rho \ge 0, \|\rho\|_1 = 1, \quad \forall p < k. \tag{152}$$

by simply choosing $\rho$ to be the point mass in any starting state.

**Optimal Policy**  We have that $\pi^\star \in \widehat{S}_j$ for all episodes $j \le k - 1$ by the inductive hypothesis. By repeatedly chaining lemma 15 for all episodes $j = 1, 2, \dots, k$ we can ensure the condition

$$\epsilon_j^{\pi^\star} \le \epsilon_j, \quad \forall j \le k. \tag{153}$$

Lemma 12 then ensures $\pi^\star \in \widehat{S}_k$.

**Empirical Optimal Policies**  We need to show that $\widehat{\pi}_{k+1}^\star \in \widehat{S}_j$ for all $j \le k$. Suppose it isn't, and let us show that this situation cannot arise. Consider the *first* episode $j \le k$ such that $\widehat{\pi}_{k+1}^\star \notin \widehat{S}_j$. Since $\widehat{\pi}_{k+1}^\star$ was in $\widehat{S}_p$, $\forall p < j$, chaining lemma 15 yields:

$$\epsilon_{j-1}^{\widehat{\pi}_{k+1}^\star} \le \epsilon_{j-1}. \tag{154}$$

The inductive hypothesis ensures[11] $\widehat{\pi}_j^\star$ was in $\widehat{S}_p$, $\forall p < j$, and so chaining lemma 15 yields:

$$\epsilon_j^{\widehat{\pi}_j^\star} \le \epsilon_j \tag{155}$$

also holds. Finally, lemma 16 ensures the following two inequalities:

$$2Cp(n_{min})\underbrace{\epsilon_{j-1}^{\widehat{\pi}_{k+1}^{\star}}}_{\leq\epsilon_{j-1}}+2Cp(n_{min})(1+C)\epsilon_{j-1}\leq\frac{\epsilon_{j-1}}{2}$$

$$2Cp(n_{min})\underbrace{\epsilon_{j}^{\widehat{\pi}_{j}^{\star}}}_{\leq\epsilon_{j}}+2Cp(n_{min})(1+C)\epsilon_{j}\leq\frac{\epsilon_{j}}{2}. \tag{156}$$

Together, equation 151, 155, 154, 156 satisfy the assumption of lemma 13 (with $\mu=\widehat{\pi}_{k+1}^{\star}$). This gives a contradiction, because the lemma claims that $\widehat{\pi}_{k+1}^{\star}$ cannot be an optimal policy while being $\notin\widehat{S}_{j}$ for some $j<k$.

The proof by induction is complete and now lemma 17 ensures

$$|(V^{\star}-\widehat{V}_{k}^{\star})(\rho)|\leq\epsilon_{k},\quad\forall\rho\geq0,\|\rho\|_{1}=1,\quad\forall k. \tag{157}$$

When the termination condition for the algorithm BESPOKE are satisfied in episode $k_{Final}$,

$$|(V^{\star}-\widehat{V}_{k_{Final}}^{\star})(\rho)|\leq\epsilon_{k_{Final}},\quad\forall\rho\geq0,\|\rho\|_{1}=1 \tag{158}$$

must hold with

$$\frac{\epsilon_{Input}}{2}\leq\epsilon_{k_{Final}}\leq\epsilon_{Input}. \tag{159}$$

Finally, the triangle inequality gives:

$$\|V^{\star}-V^{\pi_{Final}}\|_{\infty}\leq\|V^{\star}-\widehat{V}_{k_{Final}}^{\star}\|_{\infty}+\|\widehat{V}_{k_{Final}}^{\star}-V^{\pi_{Final}}\|_{\infty} \tag{160}$$

$$\leq\epsilon_{k_{Final}}+\|\widehat{V}_{k_{Final}}^{\star}-V^{\pi_{Final}}\|_{\infty} \tag{161}$$

In addition, lemma 4 ensures that for $\pi_{Final}$ at that episode:

$$\|\widehat{V}_{k_{Final}}^{\star}-V^{\pi_{Final}}\|_{\infty}\leq\epsilon_{Input}+\gamma Cp(n_{min})\|V^{\star}-V^{\pi_{Final}}\|_{\infty} \tag{162}$$

Combining the two gives:

$$\|V^{\star}-V^{\pi_{Final}}\|_{\infty}\leq\frac{2\epsilon_{Input}}{1-\gamma Cp(n_{min})}\leq2.03\epsilon_{Input} \tag{163}$$

by the choice of $n_{min}$ (see appendix A and lemma 16). $\qquad\square$

## F.2 Computational Complexity of BESPOKE

**Proposition 4** (Computational Complexity of BESPOKE). *Outside of the failure event* BESPOKE *terminates in at most*

$$\log_{2}(\frac{1}{(1-\gamma)\epsilon_{Input}})+1 \tag{164}$$

*episodes. Let $n_{sa}^{Final}$ be the total number of samples allocated by the algorithm at termination given by theorem 2. Then* BESPOKE *at termination has solved at most:*

$$(\log_{2}(\frac{1}{(1-\gamma)\epsilon_{Input}})+1)\times\log_{2}(\sum_{(s,a)}n_{sa}^{Final}) \tag{165}$$

*convex minimization programs as defined in definition 7.*

*Proof.* By the halving rule on $\epsilon_{k}$, BESPOKE must terminate after at most $\log_{2}(\frac{1}{(1-\gamma)\epsilon_{Input}})+1$ episodes; in addition, if $n_{sa}^{Final}$ is the final number sample allocated by the the algorithm to $(s,a)$ then BESPOKE solves at most $\log_{2}(\sum_{(s,a)}n_{sa}^{Final})$ convex programs as described in definition 7 at each episode. $\qquad\square$

### F.3  Sample Complexity of BESPOKE

**Theorem 2** (Sample Complexity of the Algorithm BESPOKE). *With probability at least $1 - \delta$, the total sample complexity of* BESPOKE *up to episode $k$ is upper bounded by $\sum_{(s,a)} n_{sa}$ where $n_{sa}$ is the total number of samples allocated to the $(s, a)$ pair:*

$$n_{sa} = \tilde{O} \left( \min \left\{ \frac{1}{(1-\gamma)^3 (\epsilon_k)^2}, \frac{\operatorname{Var} R(s,a) + \gamma^2 \operatorname{Var}_{p(s,a)} V^\star}{(1-\gamma)^2 (\epsilon_k)^2} + \frac{1}{(1-\gamma)^2 (\epsilon_k)}, \right. \right. \tag{166}$$

$$\left. \left. \frac{\operatorname{Var} R(s,a) + \gamma^2 \operatorname{Var}_{p(s,a)} V^\star}{\Delta_{s,a}^2} + \frac{1}{(1-\gamma)\Delta_{s,a}} \right\} + \frac{\gamma S}{(1-\gamma)^2} \right). \tag{167}$$

*Proof.* We show that a sample complexity

$$n_{sa} = \tilde{O} \left( \min \left( \underbrace{\frac{1}{(1-\gamma)^3 \epsilon_k^2}}_{A}, \underbrace{\frac{\operatorname{Var} R(s,a) + \gamma^2 \operatorname{Var}_{p(s,a)} V^\star}{(1-\gamma)^2 \epsilon_k^2} + \frac{1}{(1-\gamma)^2 \epsilon_k}}_{B_{sa}}, \right. \right. \tag{168}$$

$$\left. \left. \underbrace{\frac{\operatorname{Var} R(s,a) + \gamma^2 \operatorname{Var}_{p(s,a)} V^\star}{\Delta_{sa}^2} + \frac{1}{(1-\gamma)\Delta_{sa}}}_{C_{sa}} \right) + \frac{\gamma S}{(1-\gamma)^2} \right) \tag{169}$$

suffices to ensure that the value of the star minimax program of definition 6 is $\leq \frac{\epsilon_k}{4}$. By lemma 6 this is an upper bound on the sample complexity of the algorithm minimax program of definition 5 to guarantee that its objective is $\leq \frac{\epsilon_k}{4}$.

Three cases are possible for the $\min$ of equation 168: either the $\min$ of equation 168 is attained by $A$ or $B_{sa}$ or $C_{sa}$, and we partition the state-action space accordingly into the sets $\mathcal{A}_k, \mathcal{B}_k, \mathcal{C}_k$ corresponding to whether a certain $(s, a)$ pair attains the minimum of equation 168 with the expression $A, B_{sa}, C_{sa}$, respectively. In other words, we have the partition: $\mathcal{A}_k \cup \mathcal{B}_k \cup \mathcal{C}_k = \mathcal{S} \times \mathcal{A}$ where:

$$\mathcal{A}_k \overset{def}{=} \{(s,a) \mid A = \arg\max(A, B_{sa}, C_{sa})\} \tag{170}$$

$$\mathcal{B}_k \overset{def}{=} \{(s,a) \mid B_{sa} = \arg\max(A, B_{sa}, C_{sa})\} \tag{171}$$

$$\mathcal{C}_k \overset{def}{=} \{(s,a) \mid C_{sa} = \arg\max(A, B_{sa}, C_{sa})\} \tag{172}$$

$$\tag{173}$$

with ties broken arbitrarily.

Therefore it suffices to bound the terms below:

$$f_\star(n) = \sum_{(s,a) \in \mathcal{A}_k} \overline{w}_{sa}^{\pi,\rho} (CI_{sa}(n_{sa}^{k+1}) + 2B_{ksa}) + \sum_{(s,a) \in \mathcal{B}_k} \overline{w}_{sa}^{\pi,\rho} (CI_{sa}(n_{sa}^{k+1}) + 2B_{ksa}) \tag{174}$$

$$+ \sum_{(s,a) \in \mathcal{C}_k} \overline{w}_{sa}^{\pi,\rho} (CI_{sa}(n_{sa}^{k+1}) + 2B_{ksa}) + 23Cp(n_{min})\epsilon_k. \tag{175}$$

separately for all policies $\pi$ and starting distributions $\rho$ that satisfy:

$$(V^\star - V^\pi)(\rho) \leq C^\star \epsilon_k. \tag{176}$$

**Pairs in $\mathcal{A}_k$**  First notice that a sample complexity

$$\tilde{O} \left( \frac{1}{(1-\gamma)^2 \epsilon_k} + \frac{1}{(1-\gamma)^2} \right) \tag{177}$$

suffices to ensure

$$\sum_{(s,a) \in \mathcal{A}_k} \overline{w}_{sa}^{\pi,\rho} (2B_{ksa}) \leq \frac{\epsilon_k}{200}. \tag{178}$$

By definition of $CI_{sa}(n_{sa}^{k+1})$ we can write:

$$\sum_{(s,a)\in\mathcal{A}_k} \overline{w}_{sa}^{\pi,\rho} CI_{sa}(n_{sa}^{k+1}) = \sum_{(s,a)\in\mathcal{A}_k} \overline{w}_{sa}^{\pi,\rho}\gamma\sqrt{\frac{2\operatorname{Var}_{p(s,a)} V^\star c_n}{n_{sa}}} + \sum_{(s,a)\in\mathcal{A}_k} \overline{w}_{sa}^{\pi,\rho}\frac{\gamma c_n}{3(1-\gamma)(n_{sa}-1)}$$

$$+ \sum_{(s,a)\in\mathcal{A}_k} \overline{w}_{sa}^{\pi,\rho}\sqrt{\frac{2\operatorname{Var} R(s,a)c_n}{n_{sa}}} + \sum_{(s,a)\in\mathcal{A}_k} \overline{w}_{sa}^{\pi,\rho}\frac{c_n}{3(n_{sa}-1)} \quad (179)$$

Since $\pi$ is $C^\star\epsilon_k$ optimal for every starting distribution it must be $C^\star\epsilon_k$ optimal in the max-norm as well and hence we have the upper bound below thanks to lemma 5:

$$\leq \sum_{(s,a)\in\mathcal{A}_k} \gamma\overline{w}_{sa}^{\pi,\rho}\left(\sqrt{\frac{2\operatorname{Var}_{p(s,a)} V^\pi c_n}{n_{sa}}} + \frac{\sqrt{2c_n}}{\sqrt{n_{sa}}}C^\star\epsilon_k + \frac{c_n}{3(1-\gamma)(n_{sa}-1)}\right) \quad (180)$$

$$+ \sum_{(s,a)\in\mathcal{A}_k} \overline{w}_{sa}^{\pi,\rho}\left(\sqrt{\frac{2\operatorname{Var}_{p(s,a)} R(s,a)c_n}{n_{sa}}} + \frac{c_n}{3(n_{sa}-1)}\right) \quad (181)$$

We focus on the first term of equation 180. Cauchy-Schwartz gives:

$$\leq \frac{\gamma}{\sqrt{1-\gamma}}\sqrt{\frac{2\sum_{(s,a)\in\mathcal{A}_k} \overline{w}_{sa}^{\pi,\rho} \operatorname{Var}_{p(s,a)} V^\pi c_n}{n}} \quad (182)$$

where $n = \min_{(s,a)\in\mathcal{A}_k} n_{sa}$. Thanks to the law of total variance [AMK12] we have that $\sum_{(s,a)\in\mathcal{A}_k} \overline{w}_{sa}^{\pi,\rho} \operatorname{Var}_{p(s,a)} V^\pi$ is at most the variance of the returns upon following policy $\pi$ on the true MDP, and it is thus bounded by $1/(1-\gamma)^2$. This gives the upper bound:

$$\leq \frac{1}{(1-\gamma)^{\frac{3}{2}}}\sqrt{\frac{2c_n}{n}} \quad (183)$$

At this point a sample complexity:

$$n_{sa} = \tilde{O}\left(\frac{1}{(1-\gamma)^3\epsilon_k^2}\right) \quad (184)$$

$$n_{sa} = \tilde{O}\left(\frac{1}{(1-\gamma)^2}\right) \quad (185)$$

$$n_{sa} = \tilde{O}\left(\frac{1}{(1-\gamma)^2\epsilon_k}\right) \quad (186)$$

respectively, suffices to ensure that each term in equation 180 (i.e., the transition terms) is less than $\frac{\epsilon_k}{200}$. Since $\epsilon_k \leq \frac{1}{1-\gamma}$ we have that:

$$n_{sa} = \tilde{O}\left(\frac{1}{(1-\gamma)^3\epsilon_k^2} + \frac{1}{(1-\gamma)^2}\right) \quad (187)$$

suffices. Now we focus on the remaining terms (the reward terms), equation 181. We have the upper bound below:

$$\sum_{(s,a)\in\mathcal{A}_k} \overline{w}_{sa}^{\pi,\rho}\left(\sqrt{\frac{2c_n}{n_{sa}}} + \frac{c_n}{3(n_{sa}-1)}\right) \quad (188)$$

Again, a sample complexity of order:

$$\tilde{O}\left(\frac{1}{(1-\gamma)^2\epsilon_k^2}\right) \quad (189)$$

suffices to ensure that each term in 179 is $\leq \frac{\epsilon_k}{200}$. This implies that expression 181 is $\leq \frac{\epsilon_k}{25}$.

**Pairs in $\mathcal{B}_k$**    Notice that we have

$$\gamma\sqrt{\frac{2\operatorname{Var}_{p(s,a)} V^\star c_n}{n_{sa}}} \le (1-\gamma)\frac{\epsilon_k}{100} \tag{190}$$

$$\frac{\gamma c_n}{3(1-\gamma)(n_{sa}-1)} \le (1-\gamma)\frac{\epsilon_k}{100} \tag{191}$$

$$\sqrt{\frac{2\operatorname{Var} R(s,a)c_n}{n_{sa}}} \le (1-\gamma)\frac{\epsilon_k}{100} \tag{192}$$

$$\frac{c_n}{3(n_{sa}-1)} \le (1-\gamma)\frac{\epsilon_k}{100} \tag{193}$$

$$2B_{ksa} \le (1-\gamma)\frac{\epsilon_k}{100} \tag{194}$$

for

$$n_{sa} = \tilde{O}\left(\frac{\gamma^2 \operatorname{Var}_{p(s,a)} V^\star}{(1-\gamma)^2\epsilon_k^2}\right) \tag{195}$$

$$n_{sa} = \tilde{O}\left(\frac{\gamma}{(1-\gamma)^2\epsilon_k}\right) \tag{196}$$

$$n_{sa} = \tilde{O}\left(\frac{\operatorname{Var} R(s,a)}{(1-\gamma)^2\epsilon_k^2}\right) \tag{197}$$

$$n_{sa} = \tilde{O}\left(\frac{1}{(1-\gamma)\epsilon_k}\right) \tag{198}$$

$$n_{sa} = \tilde{O}\left(\frac{1}{(1-\gamma)^2\epsilon_k} + \frac{\gamma^2}{(1-\gamma)^2}\right) \tag{199}$$

respectively. Summing over the $(s,a)$ pairs with their maximum of type $B$ yields:

$$\sum_{(s,a)\in\mathcal{B}_k} \overline{w}_{sa}^{\pi,\rho}(CI_{sa}(n_{sa}^{k+1}) + 2B_{ksa}) \le \sum_{(s,a)\in\mathcal{B}_k} \overline{w}_{sa}^{\pi,\rho}(1-\gamma)\frac{\epsilon_k}{20} = \frac{\epsilon_k}{20}. \tag{200}$$

**Pairs in $\mathcal{C}_k$**    In this case notice that we have

$$\gamma\sqrt{\frac{2\operatorname{Var}_{p(s,a)} V^\star c_n}{n_{sa}}} \le \frac{\Delta_{sa}}{100C^\star} \tag{201}$$

$$\frac{\gamma c_n}{3(1-\gamma)(n_{sa}-1)} \le \frac{\Delta_{sa}}{100C^\star} \tag{202}$$

$$\sqrt{\frac{2\operatorname{Var} R(s,a)c_n}{n_{sa}}} \le \frac{\Delta_{sa}}{100C^\star} \tag{203}$$

$$\frac{c_n}{3(n_{sa}-1)} \le \frac{\Delta_{sa}}{100C^\star} \tag{204}$$

$$2B_{ksa} \le \frac{\Delta_{sa}}{100C^\star} \tag{205}$$

for

$$n_{sa} = \tilde{O}\left(\frac{\operatorname{Var}_{p(s,a)} V^\star}{\Delta_{sa}^2}\right) \tag{206}$$

$$n_{sa} = \tilde{O}\left(\frac{1}{(1-\gamma)\Delta_{sa}}\right) \tag{207}$$

$$n_{sa} = \tilde{O}\left(\frac{\operatorname{Var} R(s,a)}{\Delta_{sa}^2}\right) \tag{208}$$

$$n_{sa} = \tilde{O}\left(\frac{1}{\Delta_{sa}}\right) \tag{209}$$

$$n_{sa} = \tilde{O}\left(\frac{1}{(1-\gamma)\Delta_{sa}}\right) \tag{210}$$

$$\tag{211}$$

respectively. This ensures[12]

$$CI_{sa}(n_{sa}^{k+1}) + 2B_{ksa} \leq \gamma\sqrt{\frac{2\operatorname{Var}_{p(s,a)} V^\star c_n}{n_{sa}}} + \frac{\gamma c_n}{3(1-\gamma)(n_{sa}-1)} \tag{212}$$

$$+ \sqrt{\frac{2\operatorname{Var} R(s,a) c_n}{n_{sa}}} + \frac{c_n}{3(n_{sa}-1)} + \frac{\Delta_{sa}}{100C^\star} \tag{213}$$

$$\leq \frac{\Delta_{sa}}{20C^\star} \tag{214}$$

Summing over the $(s,a)$ pairs with their maximum of type $C$:

$$\sum_{(s,a)\in\mathcal{C}_k} \overline{w}_{sa}^{\pi,\rho}(CI_{sa}(n_{sa}^{k+1}) + 2B_{ksa}) \leq \frac{1}{20C^\star} \sum_{(s,a)\in\mathcal{C}_k} \overline{w}_{sa}^{\pi,\rho}\Delta_{sa} = \frac{1}{20C^\star}\left(V^\star - V^\pi\right)(\rho) \leq \frac{1}{20}\epsilon_k. \tag{215}$$

The equality arises from lemma 1 and the last inequality on the constraint on the policy for the $\star$-optimization program.

**Term** $23Cp(n_{min})\epsilon_k$   This can be made $\leq \frac{\epsilon_k}{25}$ with lemma 16 by using

$$n_{sa} = \tilde{O}\left(\frac{S}{(1-\gamma)^2}\right) \tag{216}$$

samples.

**Concluding remarks**   Summing all the upper bounds just derived for each term ensures that equation 174 is upper bounded by $\frac{\epsilon_k}{4}$ with a total sample complexity as written in the theorem statement. By lemma 6 this is an upper bound on the sample complexity of the algorithm at step $k$, and since BESPOKE reaches step $k$ after logarithmically-many episodes (see proposition 4), this is also the total sample complexity up to episode $k$ up to log factors.

$\square$

# G   Efficient Implementation

In this section we rewrite the minimax optimization program of definition 5 (with $\epsilon_k^\pi = \epsilon_k$) into a convex minimization program that can be efficiently solved. First we directly optimize over the distribution[13] $w$ instead of over the policy $\pi$ and introduce an appropriate scalar slack variable $t$. This

allows us to put the inner maximization in the following matrix form (we neglect the constant term $+2Cp(n_{min})\epsilon$):

$$
\begin{aligned}
\max_{x} \quad & c^\top x \\
\text{s.t.} \quad & Ax = b \\
& x \geq 0
\end{aligned}
\tag{217}
$$

with

$$
c = \begin{bmatrix} \widehat{CI}^{k+1} + B_k \\ 0 \\ 0 \end{bmatrix}; \quad x = \begin{bmatrix} w \\ \rho \\ t \end{bmatrix}; \quad A = \begin{bmatrix} \Xi - \gamma \widehat{P}_k^\top & -I & 0 \\ 0 & \mathbb{1}^\top & 0 \\ \widehat{r}_k^\top & -\widehat{V}_k^\star & -1 \end{bmatrix}; \quad b = \begin{bmatrix} 0 \\ 1 \\ -C\epsilon_k \end{bmatrix}.
$$

Here $I$ is the identity and $\Xi$ is a marginalization matrix as described in [WBS07]. Written explicitly, we have:

$$
[\widehat{CI}^{k+1} + B_k]_{sa} = \left( \sqrt{2c_n \operatorname{Var}\left(\widehat{R}(s,a)\right)} + \gamma \sqrt{2c_n \operatorname*{Var}_{\widehat{p}_k(s,a)} \widehat{V}_k^\star} + \gamma \sqrt{2c_n} \epsilon_k \right) \left( \frac{1}{\sqrt{n_{sa}}} \right)
\tag{218}
$$

$$
+ \left( \frac{7c_n}{3(1-\gamma)} \right) \left( \frac{1}{n_{sa} - 1} \right)
\tag{219}
$$

Notice that the above expression is a convex function of the $n_{sa}$ for $n_{sa} \geq 2$.

We compute the dual of the linear program above (with $n_{sa}$ fixed):

$$
\begin{aligned}
\min_{y} \quad & b^\top y \\
\text{s.t.} \quad & A^\top y \geq c
\end{aligned}
\tag{220}
$$

Therefore, the minimax program of definition 5 can be reformulated into an equivalent convex minimization program (now we add back $+2Cp(n_{min})\epsilon$ and the outer minimization program):

**Definition 7** (Convex Minimization Program).

$$
\begin{aligned}
\min_{n,y} \quad & b^\top y + 2Cp(n_{min})\epsilon \\
\text{s.t.} \quad & c - A^\top y \leq 0 \\
& n_{sa} \geq 0, \quad \forall (s,a) \\
& \sum_{(s,a)} n_{sa} \leq n_{max}.
\end{aligned}
\tag{221}
$$