[Reviews · NeurIPS 2019]

Reviewer 1



Designing sample-efficient RL algorithms is a fundamental open problem and this paper makes a good step in such direction. Though the case where a generative model is available is limited with respect to the general online case, it is still important to fully understand the intrinsic complexity of solving MDPs. Furthermore, exploiting structure to reduce sample complexity is an important active area of research and has not been fully analyzed for the case where a generative model is available. This work presents interesting and novel ideas in deriving the proposed algorithm together with strong and significant theoretical results. The paper reads well and it is easy to follow. Although the proofs are quite long, the authors successfully present the main intuitions in the main paper while deferring all technical details to the appendix. The latter is well organized (in particular, the table of all symbols in Appendix A is really helpful), which facilitates proofreading. I went through all the proofs and they seem to be ok (I have only a couple of questions, see below). Overall I think this is a good paper and I vote for acceptance. I have only some minor comments/questions: 1. Section 6 discusses the relation between the sample complexity bound of Th. 2 and the existing worst-case lower/upper bounds. Since the worst-case instance that is created in the derivation of the existing lower-bound does not reflect the problem structure considered here, have you thought about deriving a problem-dependent lower bound for this problem? 2. Some empirical simulations would strengthen even more the theoretical results presented in the paper 3. In the derivation of Eq. 60, how did you get rid of the terms \sum_{sa} \hat{w}_{sa}B_{ksa}? After appyling Lemma 12 and summing and subtracting \sum_{sa} \bar{w}_{sa}(CI_{sa} + 2B_{ksa}), we can use Lemma 10 to get rid of \sum_{sa} (\hat{w}_{sa} - \bar{w}_{sa})CI_{sa}, but what about \sum_{sa} (\hat{w}_{sa} - \bar{w}_{sa})2B_{ksa}? 4. I am not sure how the second statement of Th. 1 (||V^* - V^{\pi_k}|| \leq 2\epsilon) is proved. At the end of the proof, using the triangle inequality, I obtain: ||V^* - V^{\pi_k}|| \leq ||V^* - V_k^*|| + ||V_k^* - V^{\pi_k}||. Using Eq. 152, the first term is ||V^* - V_k^*|| \leq \epsilon_k. Using Lemma 5, the second term is ||V_k^* - V^{\pi_k}|| \leq \epsilon_k^{\pi_k}+ Cp(n_min)||V^* - V^{\pi_k}||. Rearranging, ||V^* - V^{\pi_k}|| \leq 2\epsilon_k/(1 - Cp(n_min)). Am I missing something? Some minor typos I have found: - line 29: brings - line 32: improves and worst-case - line 76: delta was not defined (though most readers know what that is) - line 77: the failure event was not defined (and the confidence intervals needed to define it were not defined either) - line 133-134: samples is repeated twice - in Alg. 1: should't \epsilon_0 be \epsilon_1 (the index k starts at 1). There is also a reference to Def. 7 (not in the main paper). Is it Def. 2? - line 187: to be able - line 203: Lemma 1 was already used in Sec. 2 - Eq. 32: shouldn't there be a minus? --- Post-rebuttal update: The authors successfully corrected the two minor errors that I had spotted in my review (only numerical constants were affected anyway). I am now rather confident that the proofs are correct. I thus confirm my initial view and vote for acceptance. All reviewers suggested some numerical simulations as a direction for possible improvement, but I think this is of minor importance given the strong theoretical results. Anyway, should the authors complete some experiments by the camera-ready deadline, I suggest using the extra page to include them and/or to better clarify the points that were not clear to the other reviewers.

Reviewer 2



Originality: It is a new method compared with previous bound where the paper use a non-uniform sampling scheme. Quality: The paper provides detailed theoretical analysis on their methods. However, no empirical experiment is provided. It would be better even if a toy example is provided and can show significant improvement of sample complexity than previous methods. Clarity: I'm a little bit struggling on the writing. The paper is notation heavy and hard to check all the proofs. I would suggest to emphasize on the notation of $n_{sa}^k$ which I think is your key differences than previous methods, should spend more paragraph on this notation. I didn't understand the intuition why problem dependent structure can remove the dependence of horizon for suboptimal action. Another thing is the terminology for 'horizon' where we usually use to refer to the maximum time steps on the MDP. In this paper it refers to the factor $(1-\gamma)^{-1}$ which is not fully explained. Significance: To be honest I don't familiar with the field of PAC-RL, but I believe it is an important result and can inspire policy optimization algorithm based on your algorithm. Overall: I believe the paper can be written much better by emphasizing its idea and explain better on its intuition. But I still think this paper is above borderline because of its theoretical quality and the idea of non-uniform sample would be inspiring for other policy optimization algorithm.

Reviewer 3



The authors provide a new sample complexity analysis for identifying a near optimal policy of a tabular MDP assuming access to a generative model. They provide a problem dependent bound compared to the worst case bounds present in the literature. The analysis technique is novel in the sense that their bounds depend on the gaps between the optimal action value function and value function of suboptimal actions. Similar bounds are well known in the multi armed bandits literature and shown to be derived from the proposed bound. I am not much familiar with the relevant RL literature and hence can not make detailed comments. The bound improves over the previous results and a high level proof sketch is present in the paper. However, the paper refers to several lemmas present in the supplementary material which makes it a difficult read. Given the short amount of time I was not able to go through the supplementary material. But I assume that the proofs are correct. One novel idea behind the proof is to make use of the variance of the optimal value function under the next state distribution, which effectively helps to improve over the previous analysis. I believe this new technique might be helpful to the community. --------------- Post rebuttal update: The authors mentioned some ideas about numerically evaluating their methods. I confirm my original view and vote for acceptance.

[Author Response · NeurIPS 2019]

1  The authors thank the reviewers for their helpful comments.

2  **=== R1 ===**

3  **1)** We have indeed thought about a lower bound, but do not yet have a full result yet. The contribution for the suboptimal
arms is essentially tight (excluding logs and constants) from the bandit literature. However, for the sample complexity
along the optimal state-actions we are not sure: the extra $1/(1-\gamma)$ factor in the upper bound, which is in some way
unavoidable due to the worst case lower bound, stems from the discounted sum of visit probabilities, which might
interact in a non-trivial way with the variances in constructing a lower bound. Finally, the constant term $S/(1-\gamma)^2$
is likely to be avoidable: a paper titled "On the Optimality of Sparse Model-Based Planning for Markov Decision
Processes" that just appeared on Arxiv shows how to reduce that dependence for small $\epsilon$ in model-based approaches
like ours, and their technique seems applicable to our case. **3)** We thank the reviewer for the careful reading of the
proofs. The reviewer helpfully identified two small errors, that only impact the numerical constants. The first is in
Equation 60 where there is indeed a lower order term that increases the numerical constant. By definition of $B_{ksa}$ in
Appendix A we have:

$$\sum_{(s,a)} \left( \overline{w}_{sa}^{\pi,\rho} - \widehat{\overline{w}}_{sa}^{\pi,k,\rho} \right)\left( 2B_{sa}^k \right) = 2\sum_{(s,a)} \left( \overline{w}_{sa}^{\pi,\rho} - \widehat{\overline{w}}_{sa}^{\pi,k,\rho} \right)\left( \frac{2c_n}{(1-\gamma)(n_{sa}-1)} + \gamma\sqrt{\frac{2c_n}{n_{sa}}}\epsilon_k \right)$$

14  We examine the two different terms. For the first term using the definition of $CI_{sa}^k$ and lemma 10:

$$2\sum_{(s,a)} \left( \overline{w}_{sa}^{\pi,\rho} - \widehat{\overline{w}}_{sa}^{\pi,k,\rho} \right)\frac{2c_n}{(1-\gamma)(n_{sa}-1)} = 2\sum_{(s,a)} \left( \overline{w}_{sa}^{\pi,\rho} - \widehat{\overline{w}}_{sa}^{\pi,k,\rho} \right)\left( 6\times\underbrace{\frac{c_n}{3(1-\gamma)(n_{sa}-1)}}_{\leq CI_{sa}^k} \right) \leq 12\sum_{(s,a)} \left( \overline{w}_{sa}^{\pi,\rho} - \widehat{\overline{w}}_{sa}^{\pi,k,\rho} \right)CI_{sa}^k \leq 12Cp(n_{min})\epsilon_k^{\pi}$$

15  and for the second term using the definition of $Cp(n_{min})$ in Appendix A and $\sum_{(s,a)} \left( \overline{w}_{sa}^{\pi,\rho} + \widehat{\overline{w}}_{sa}^{\pi,k,\rho} \right) = 2/(1-\gamma)$:

$$2\sum_{(s,a)} \left( \overline{w}_{sa}^{\pi,\rho} - \widehat{\overline{w}}_{sa}^{\pi,k,\rho} \right)\left( \underbrace{\gamma\sqrt{\frac{2c_n}{n_{sa}}}}_{\leq(1-\gamma)Cp(n_{min})}\epsilon_k \right) \leq 4\sum_{(s,a)} \left( \overline{w}_{sa}^{\pi,\rho} + \widehat{\overline{w}}_{sa}^{\pi,k,\rho} \right)(1-\gamma)Cp(n_{min})\epsilon_k = 8Cp(n_{min})\epsilon_k$$

16  In summary, equation (60) in the paper would become:

$$\sum_{(s,a)} \overline{w}_{sa}^{\pi,\rho}(CI_{sa}(n_{sa}^{k+1}) + 2B_{ksa}) + 15Cp(n_{min})\epsilon_k^{\pi} + 8Cp(n_{min})\epsilon_k$$

17  This way the rest of the proofs remain unchanged in the appendix (since we show $\epsilon_k^{\pi} \leq \epsilon_k$ for the good policies), and
the numerical constants can be incorporated into the $\tilde{O}$ notation. There are ways that avoid increasing the constants as
we showed above, but the argument was not self-contained enough to be explained in the rebuttal. **4)** The reviewer
is again correct in obtaining $\|V^\star - V^{\pi_k}\|_\infty \leq 2\epsilon_k/(1-Cp(n_{min}))$ as a final bound. Lemma 18 gives a value for
$Cp(n_{min}) \leq 1/100$ and hence $\|V^\star - V^{\pi_k}\|_\infty \leq 2.03\epsilon_k$, instead of the reported $\|V^\star - V^{\pi_k}\|_\infty \leq 2\epsilon_k$.

22  **===R3===** The notation $n_{sa}^k$ represents the number of samples allocated in the $k$th phase of the algorithm to the $(s,a)$
pair. For "why problem dependent structure can remove the dependence of horizon for suboptimal action", as the
reviewer notes, the approach is quite technical and we will strive to better convey the intuition for why suboptimal
actions do not require as many samples. The key ideas are in Fact 1 and Lemma 1, where we highlight how suboptimality
depends on the distribution of visited state-action pairs, and how by adaptively allocating samples, in a way that depends
on the gaps, we can avoid a horizon dependence for suboptimal actions. Finally, some authors do 'heuristic' translations
of sample complexity between finite horizon and infinite horizon, where the number of steps for the finite case is
roughly translated into the $1/(1-\gamma)$ factor, but as the reviewer points out, we need to define the word 'horizon' for our
submission.

31  **===R4===** The authors understand that the main suggestion for improvement are a method for computing the maximum
likelihood MDP. We can report the maximum likelihood formulas for the rewards and transition probabilities in the
appendix; after this, an algorithm like policy iteration (applied to the MDP with the computed maximum likelihood
rewards and transition probabilities) can give the empirically optimal policy and value function.

35  **=== Numerical Experiments R1, R3, R4 ===** We have worked towards an implementation to answer the reviewers'
request of providing experiments, but unfortunately we did not complete it in time for the rebuttal. In addition, obtaining
an implementation that takes full advantage of our problem dependent analysis involves a more careful computation of
the numerical constants (which matter in practice) and using the law of iterated logarithms to lower the log dependence to
log-log (this is standard practice to improve the practical performance of algorithms based on concentration inequalities).

[Meta-Review · NeurIPS 2019]

All the reviewers agree that the submission makes valuable algorithmic and theoretical contributions to learning (near) optimal policies for MDPs assuming a generative model is available. Thus, I am glad to recommend the paper for acceptance.